# Learning Energy-Based Generative Models via Coarse-to-Fine Expanding and Sampling

**Yang Zhao, Jianwen Xie, Ping Li**

Cognitive Computing Lab
Baidu Research
10900 NE 8th St. Bellevue, WA 98004, USA
{yangzhao.eric, jianwen.kenny, pingli98}@gmail.com

## Abstract

Energy-based models (EBMs) parameterized by neural networks can be trained by the Markov chain Monte Carlo (MCMC) sampling-based maximum likelihood estimation. Despite the recent significant success of EBMs in image generation, the current approaches to train EBMs are unstable and have difficulty synthesizing diverse and high-fidelity images. In this paper, we propose to train EBMs via a multistage coarse-to-fine expanding and sampling strategy, which starts with learning a coarse-level EBM from images at low resolution and then gradually transits to learn a finer-level EBM from images at higher resolution by expanding the energy function as the learning progresses. The proposed framework is computationally efficient with smooth learning and sampling. It achieves the best performance on image generation amongst all EBMs and is the first successful EBM to synthesize high-fidelity images at $512 \times 512$ resolution. It can also be useful for image restoration and out-of-distribution detection. Lastly, the proposed framework is further generalized to the one-sided unsupervised image-to-image translation and beats baseline methods in terms of model size and training budget. We also present a gradient-based generative saliency method to interpret the translation dynamics.

## 1 Introduction

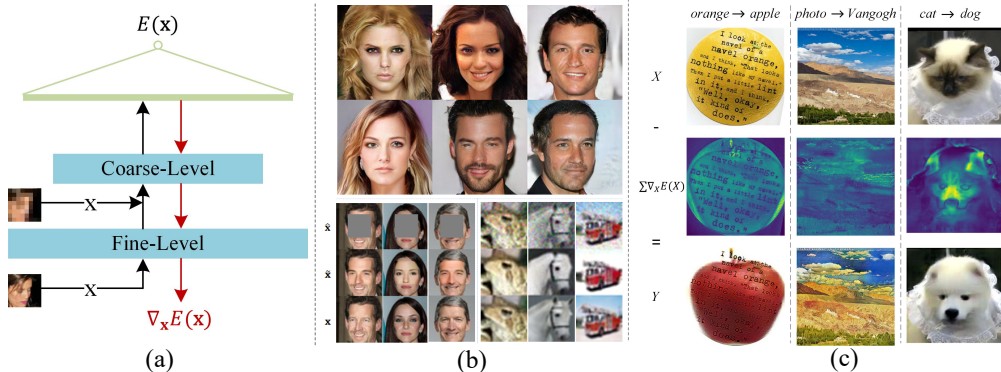

Figure 1: Highlights of our contributions. (a) Coarse-to-fine EBM with gradient-based MCMC sampling. (b) High-fidelity image generation ($512 \times 512$), image denoising, and image inpainting from top to bottom. ($\hat{x}$: corrupted image, $\tilde{x}$: restored image, $x$: ground truth). (c) Energy-based unsupervised image-to-image translation ($256 \times 256$) interpreted by the generative saliency.

Recently, energy-based models (EBMs) (Zhu et al., 1998; LeCun et al., 2006) parameterized by modern neural networks have drawn much attention from the deep learning communities. Successful applications with EBMs include generations of images (Xie et al., 2016; 2018b; Du & Mordatch, 2019), videos (Xie et al., 2017; 2019), 3D volumetric shapes (Xie et al., 2018c; 2020), unordered point clouds (Xie et al., 2021a), texts (Deng et al., 2020), molecules (Ingraham et al., 2018; Du

et al., 2019), etc., as well as image-to-image translation (Xie et al., 2021b;c), out-of-distribution detection (Liu et al., 2020) and inverse optimal control (Xu et al., 2019). EBMs are characterized by (*i*) Simplicity: The maximum likelihood learning of EBMs unifies representation and generation in a single model, and (*ii*) Explicitness: EBMs provide an explicit density distribution of data by training an energy function that assigns lower values to observed data and higher values to unobserved ones.

However, it is still difficult to train an EBM to synthesize diverse and high-fidelity images. The maximum likelihood estimation (MLE) of EBMs requires the Markov chain Monte Carlo (MCMC) (Liu, 2008; Barbu & Zhu, 2020) to sample from the model and then updates the model parameters according to the difference between those samples and the observed data. Such an "analysis by synthesis" (Grenander et al., 2007) learning scheme is challenging because the sampling step is neither efficient nor stable. In particular, when the energy function is multimodal due to the highly varied or high resolution training data, it is not easy for the MCMC chains to traverse the modes of the learned model. Fortunately, it is common knowledge that the manifold residing in a downsampled low-dimensional image space is smoother than that in the original high-dimensional counterpart. Thus, learning an EBM from low-dimensional data is much stabler and faster than learning from high-dimensional data in terms of convergence (Odena et al., 2017; Gao et al., 2018).

Inspired by the above knowledge, we propose to train EBMs via a multistage coarse-to-fine expanding and sampling strategy (CF-EBM). As shown in Figure 1(a), the approach starts with learning a coarse-level EBM on low resolution images and then smoothly transits to learn the finer-level EBM by adding new layers that take into account the higher resolution information as the learning progresses. The gradient-based short-run MCMC (Nijkamp et al., 2019), *e.g.*, Langevin dynamics (Neal et al., 2011), is used for sampling. From the modeling aspect, the coarse-level training can be useful for exploring the global structure of image, while the fine-level training will then gradually refine the image details. Recent works have demonstrated the advantages of this incremental learning (Karras et al., 2018; Wang et al., 2018). However, there have been no works focusing on the incremental learning of EBMs that incorporates bottom-up representation and top-down sampling in a single net. Besides, as shown in Figure 1(a), the top-down gradient information for synthesis flows from coarse-level layers towards fine-level layers. Thus, during the coarse-to-fine expanding, we can use the coarse-level synthesis to help the fine-level synthesis to stabilize the sampling. Such a coarse-to-fine expanding and sampling scheme is useful for high-fidelity synthesis in several vision tasks. See Figure 1(b).

Furthermore, we propose a one-sided energy-based unsupervised image-to-image translation method and scale it up to high resolution. The approach is immediately available with the FC-EBM by using its iterative Langevin dynamics without the need of the cycle consistency (Zhu et al., 2017) or geometry constraints (Fu et al., 2019). Specifically, we learn an EBM of target domain with Langevin dynamics initialized by the examples from source domain. The resulting translator is the short-run MCMC. Compared with those prior works (Zhu et al., 2017; Huang et al., 2018; Park et al., 2020) that learn black-box encoder-decoder networks between domains, our method is much more interpretable in the sense that ours can be explained by a visualization method (Simonyan et al., 2014; Adebayo et al., 2018) that uses gradients to visualize the most essential regions, *i.e.*, the generative saliency, when translating an image from the source domain to the target domain. See Figure 1(c).

The contributions of our paper can be summarized as:

- To the best of our knowledge, this is the first work that trains EBMs under the "analysis by synthesis" scheme via a multistage coarse-to-fine expanding and sampling strategy. Besides, we propose several essential techniques for improving EBM, *e.g.*, smooth activations. Particularly, our work is the first to train a pure EBM for synthesizing $512 \times 512$ images.

- We propose a novel energy-based unsupervised image-to-image translation approach, which is essentially different from all other existing GAN-based approaches. We demonstrate noticeable results in terms of both translation quality and efficiency of time and memory.

- We conduct extensive experiments to validate our approach, including image generation, denoising, inpainting, out-of-distribution detection and unsupervised image translation. Strong results show that our method outperforms or is competitive with the prior art.

The rest of the paper is organized as follows. Section 2 summarizes related works and how the proposed method is different from the prior art. Section 3 introduces the proposed methodology in detail. Section 4 presents extensive experiments to test our method. Section 5 concludes our paper and discuss some future research directions.

## 2 RELATED WORK

### 2.1 ENERGY-BASED GENERATIVE MODELS

The main challenge to train EBMs via MLE lies in drawing fair samples from the model, especially when the energy function is parameterized by a highly non-linear ConvNet. The contrastive divergence (CD) (Hinton, 2002; Tieleman, 2008), with MCMC chains initialized from data distribution, is an efficient but biased way to train EBMs. Another direction is to adopt the idea of energy-based correction of a more tractable model to train EBMs. Noise contrastive estimation (NCE) (Gutmann & Hyvärinen, 2010; Gao et al., 2020) and introspective neural networks (INNs) (Lazarow et al., 2017; Jin et al., 2017; Lee et al., 2018b) belong to this theme. Generative cooperative networks (CoopNets) (Xie et al., 2018b; 2021d;b) train an EBM with a generator or a variational auto-encoder (VAE) (Kingma & Welling, 2014) as amortized sampler by MCMC teaching (Xie et al., 2018a). Triangle divergence (Han et al., 2019) trains an EBM without MCMC by amortizing the MCMC via a VAE. However, these frameworks still struggle to scale up and model multimodal data. There have been several strategies to improve the EBM training. Gao et al. (2018) adopts a multi-grid method that trains multiple EBMs at different grids simultaneously, where the EBM at coarser grid is used to initialize the image generation by EBM at finer grid. However, optimizing and sampling from multiple EBMs will result in low efficiency of both time and memory. To stabilize the training, Nijkamp et al. (2019); Grathwohl et al. (2020) add the Gaussian white noise to the observed data, resulting in noisy synthesized images. In contrast, our paper proposes to train a single EBM via a coarse-to-fine growing strategy, along with some improved techniques. With smooth parameter training and image sampling, our model can preserve EBM's compatibility and synthesize high-fidelity images.

### 2.2 UNSUPERVISED IMAGE-TO-IMAGE TRANSLATION

Unsupervised image-to-image translation aims at learning two directions of mappings between two unpaired domains. Recent successes are all based on adversarial learning, *e.g.*, CycleGAN (Zhu et al., 2017), UNIT (Liu et al., 2017), MUNIT (Huang et al., 2018), DRIT (Lee et al., 2018a) and U-GAT-IT (Kim et al., 2020). These methods typically train two GANs with two levels of learning objectives: (*i*) Distribution level: Two adversarial losses are used to capture style discrepancy between source and target domain; (*ii*) Instance level: To tackle the difficulty of unpaired setting, they adopt a cycle consistency loss for content preservation. This loss enables an instance-level supervision to regularize the training of two mappings by enforcing them to be a bijective function between two domains. Except for works about two-sided translation, efforts on research about one-sided unsupervised image translation have also been made, *e.g.*, DistanceGAN (Benaim & Wolf, 2017), GcGAN (Fu et al., 2019) and CUT (Park et al., 2020), which apply geometric or contrastive constraints. We solve this problem from the prospective of EBM, which is different from GAN-based methods. The proposed concise EBM solution only relies on its built-in objective, which is a distribution-level statistics matching, to accomplish the one-sided image translation. It transfers the style and preserves the source content by MCMC without using the cycle-consistency loss. The model demonstrates better performances with less time and memory. Another distinction between our method and GAN-based methods is the natural interpretability of Langevin dynamics. It provides a gradient-based saliency map to visualize those key regions that make the two domains distinct, as illustrated in Figure 1(c).

## 3 METHOD

In this section, we first present the EBM learning framework, and then the proposed CF-EBM approach. After that, we generalize our model to the unsupervised image-to-image translation.

### 3.1 MCMC-BASED MAXIMUM LIKELIHOOD LEARNING OF ENERGY-BASED MODEL

Let $\mathbf{x} \in \mathbb{R}^D$ be the observed example, *e.g.*, an image. An energy-based model is defined as follows:

$$p_\theta(\mathbf{x}) = \frac{1}{Z(\theta)} \exp(-E_\theta(\mathbf{x})), \tag{1}$$

where $E_\theta(\mathbf{x}) : \mathbb{R}^D \to \mathbb{R}$ is the energy function defined by a bottom-up ConvNet parameterized by $\theta$. $Z(\theta) = \int \exp(-E_\theta(\mathbf{x}))d\mathbf{x}$ is the intractable normalizing constant or the partition function. Given $N$ observed examples $\{\mathbf{x}_i\}_{i=1}^N \sim p_{\text{data}}(\mathbf{x})$, where $p_{\text{data}}(\mathbf{x})$ denotes the unknown data distribution, the model can be trained by maximizing the log-likelihood $L(\theta) = \frac{1}{N}\sum_{i=1}^N \log p_\theta(\mathbf{x}_i) \approx$

$\mathbb{E}_{\mathbf{x} \sim p_{\text{data}}(\mathbf{x})} \log(p_\theta(\mathbf{x}))$. The derivative of the negative log-likelihood is given by

$$-\nabla_\theta L(\theta) = \mathbb{E}_{\mathbf{x} \sim p_{\text{data}}(\mathbf{x})}[\nabla_\theta E_\theta(\mathbf{x})] - \mathbb{E}_{\tilde{\mathbf{x}} \sim p_\theta(\mathbf{x})}[\nabla_\theta E_\theta(\tilde{\mathbf{x}})], \qquad (2)$$

where the second expectation term under $p_\theta(\mathbf{x})$ is intractable and can be approximated via MCMC. Given that, the EBM is updated by gradient descent. To sample $\tilde{\mathbf{x}} \sim p_\theta(\mathbf{x})$ via MCMC, we rely on gradient-based Langevin dynamics that recursively computes the following step

$$\tilde{\mathbf{x}}_{t+1} = \tilde{\mathbf{x}}_t - \frac{\eta_t}{2} \nabla_{\tilde{\mathbf{x}}} E_\theta(\tilde{\mathbf{x}}_t) + \sqrt{\eta_t}\epsilon_t, \quad \epsilon_t \sim \mathcal{N}(0, \mathbf{I}), \qquad (3)$$

where $\eta_t$ is the step size of Langevin step and also the variance of Gaussian noise $\epsilon_t$. Theoretically, to ensure convergence, the MCMC is typically performed with infinite steps and an infinitesimal stepsize (Welling & Teh, 2011). However, it is impractical for training EBMs. In this paper, we follow Nijkamp et al. (2019) to use short-run MCMC, which always starts from a fixed noise distribution and runs a fixed number $T$ of Langevin steps in both training and testing stages. The training with a short-run MCMC might result in a biased estimation of EBM but the learned short-run MCMC is still a valid generator, which enables us to synthesize realistic images and efficiently train the model, as seen in most well-established EBM works (Nijkamp et al., 2019; Grathwohl et al., 2020; Pang et al., 2020). In this paper, we keep the step size constant and linearly decay the noise variance till 0.

## 3.2 CF-EBM: Coarse-to-fine Expanding and Sampling of Energy-Based Model

Our primary contribution is the multistage coarse-to-fine expanding and sampling methodology for training EBMs. The key idea is to incrementally grow the EBM from a low resolution (coarse model) to a high resolution (fine model) by gradually adding new layers to the energy function. In this way, both stability and time efficiency in training EBMs benefit. And eventually, we only keep the EBM at the highest resolution for image generation using the short-run MCMC sampling.

Figure 2 (a) illustrates the proposed multistage coarse-to-fine EBM training strategy, which is accompanied by the pseudocode in Algorithm 1. Let $S$ denote the total number of training stages and $(\mathbf{x}^{(s)}, s = 1, ..., S)$ denote the multi-resolution versions of an image $\mathbf{x}$, with $\mathbf{x}^{(0)}$ being the minimal resolution version of $\mathbf{x}$, and $\mathbf{x}^{(S)} = \mathbf{x}$. For each $\mathbf{x}^{(s)}$, we can easily generate $\mathbf{x}^{(s-1)}$ by average pooling with a pool-

---

**Algorithm 1:** CF-EBM Training

**Input:** Multi-resolution data $\{\mathbf{x}_i^{(s)}, i = 1, ..., N; s = 1, ..., S\}$, numbers of Langevin steps $\{T^{(s)}\}_{s=1}^S$, and batch sizes $\{n^{(s)}\}_{s=1}^S$
**Output:** Energy function $E^{(S)}$

$E^{(0)} \leftarrow \varnothing$
**for** $s = 1 : S$ **do**
    $m = 0, \ \beta = 0$
    Expand model: $E^{(s)} \leftarrow E^{(s-1)} + \texttt{Expand}(\cdot)$
    **while** $(m \le N)$ **do**
        # Draw real data and initial MCMC samples
        Draw observed $\{\mathbf{x}_i^{(s)}\}_{i=1}^{n^{(s)}} \sim p_{\text{data}}(\mathbf{x})$
        Draw initial $\{\tilde{\mathbf{x}}_i^{(s)}\}_{i=1}^{n^{(s)}} \sim \mathcal{U}^{(s)}(-1, 1)$
        # Smooth MCMC sampling
        **if** $(\beta < 1 \text{ and } s > 1)$ **then**
            Draw initial $\{\tilde{\mathbf{x}}_i^{(s-1)}\}_{i=1}^{n^{(s)}} \sim \mathcal{U}^{(s-1)}(-1, 1)$
            Update $\{\tilde{\mathbf{x}}_i^{(s-1)}\}_{i=1}^{n^{(s)}}$ for $T^{(s-1)}(1 - \beta)$ steps of Eq. (3) with $E^{(s-1)}$
            $\tilde{\mathbf{x}}_i^{(s)} \leftarrow \sqrt{1 - \beta^2}\texttt{Upsample}(\tilde{\mathbf{x}}_i^{(s-1)}) + \beta\tilde{\mathbf{x}}_i^{(s)}$
        **end**
        Update $\{\tilde{\mathbf{x}}_i^{(s)}\}_{i=1}^{n^{(s)}}$ for $T^{(s)}$ steps of Eq. (3) with $E^{(s)}$
        # Optimize energy function
        Update $E^{(s)}$ with gradient descent in Eq. (2)
        $m \leftarrow m + n^{(s)}$
        $\beta = \max(1, \ 2N/n^{(s)})$
    **end**
**end**

---

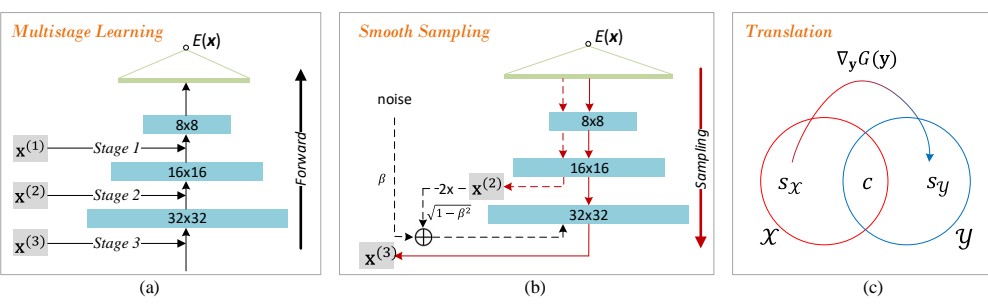

Figure 2: Illustrations of (a) multistage CF-EBM, (b) smooth sampling, and (c) image translation.

ing size. At stage $s$, we associate the model with an energy function $E^{(s)}$ to learn from $\mathbf{x}^{(s)}$. The model starts from an initial minimal resolution at stage 1. When the training proceeds to the next stage $s + 1$, we will add randomly initialized layers at the bottom to expand the resolution, and obtain a new energy function $E^{(s+1)}$. For example, in Figure 2 (a), when the training stage switches from stage 2 to 3, we expand $E^{(2)}$, which is an energy function of resolution $16 \times 16$, by adding a bottom block accounting for resolution $32 \times 32$ to get $E^{(3)}$. After that, we only need to train $E^{(3)}$ at stage 3.

Compared with PGAN (Karras et al., 2018) which proposes a progressive growing strategy to train GAN, our CF-EBM integrates both the learning (Eq. (2)) and the sampling (Eq. (3)) into a single neural network $E_\theta(\mathbf{x})$, which is essentially different from PGAN that applies a discriminator to guide the learning and a generator to produce samples. Therefore, training an EBM via a multistage growing strategy is more challenging and non-trivial. With the newly expanded energy function $E^{(s+1)}$ and the enlarged training examples $\mathbf{x}^{(s+1)}$, the stage transition must be taken care of to avoid instability issue. The CF-EBM deals with this issue from the following two aspects:

**(i) Smooth EBM Learning:** Let $\texttt{Expand}(\cdot)$ denote an expanding block. To expand the resolution of the EBM at each stage, we add an $\texttt{Expand}(\cdot)$ to the bottom of the energy function to double its resolution. We fade in the block smoothly to prevent sudden shocks to the well-trained coarse-level energy function. Specifically, $\texttt{Expand}(\cdot)$ is a composition of a primal block $\texttt{Primal}(\cdot)$ and a fading block $\texttt{Fade}(\cdot)$. We define $\texttt{Expand}(\mathbf{x}) = \beta \texttt{Primal}(\mathbf{x}) + (1 - \beta)\texttt{Fade}(\mathbf{x})$, where $\beta$ is a fading factor that gradually increases from 0 to 1 as more data are consumed in the training. As a result, the model will first rely on the well-trained coarse-level layers and gradually shift the attention to the newly added fine-level layers. The auxiliary fading block will be abandoned when $\beta$ reaches 1. In this paper, $\texttt{Primal}(\cdot)$ consists of two convolution layers and $\texttt{Fade}(\cdot)$ is a convolution layer followed by a $2 \times 2$ average pooling to link the previous coarse-level layers.

**(ii) Smooth MCMC Sampling:** The smooth sampling is realized both implicitly and explicitly.

($a$) *Implicitly*: Considering the resolution transition from stage $s - 1$ to stage $s$, we expand the energy function by $E^{(s)}(\mathbf{x}^{(s)}) = E^{(s-1)}(\texttt{Expand}(\mathbf{x}^{(s)}))$. The gradient $\nabla_{\mathbf{x}^{(s)}} E^{(s)}(\mathbf{x}^{(s)})$ for Langevin sampling in Eq. (3) can be unfolded as:

$$\nabla_{\texttt{Expand}(\mathbf{x}^{(s)})} E^{(s-1)}(\texttt{Expand}(\mathbf{x}^{(s)})) \times [\beta \nabla_{\mathbf{x}^{(s)}} \texttt{Primal}(\mathbf{x}^{(s)}) + (1 - \beta)\nabla_{\mathbf{x}^{(s)}} \texttt{Fade}(\mathbf{x}^{(s)})].$$

Initially, because the fading factor $\beta$ is small, the newly added primal block $\texttt{Primal}(\cdot)$, whose weights are randomly initialized, would not largely affect the sampling immediately. Only the coarse-level function $E^{(s-1)}$ and the auxiliary fading block $\texttt{Fade}(\cdot)$ make major contributions to the image synthesis. As $\beta$ increases, the $\texttt{Primal}(\mathbf{x}^{(s)})$ becomes increasingly well-trained. Meanwhile, the term $\nabla_{\mathbf{x}^{(s)}} \texttt{Fade}(\mathbf{x}^{(s)})$ fades away and $\nabla_{\mathbf{x}^{(s)}} \texttt{Primal}(\mathbf{x}^{(s)})$ gradually takes the lead in the sampling, thus the synthesized images would become sharper and sharper.

($b$) *Explicitly*: To mitigate the impact of the sudden model expansion on sampling, we utilize the well-trained coarse model $E^{(s-1)}$ to initialize the MCMC sampling of the newly expanded model $E^{(s)}$. Specifically, at stage $s$, the algorithm first generates low resolution samples from $E^{(s-1)}$ by running $T^{(s-1)}(1 - \beta)$ Langevin steps. Those samples are then upsampled ($2\times$) by $\texttt{Upsample}(\cdot)$ and mixed with a uniform noise to initialize the MCMC sampling of $E^{(s)}$. As the fading factor $\beta$ increases, such an MCMC initialization assistance from the coarse model would fade away. Eventually, when $\beta$ reaches 1, we can directly sample from $E^{(s)}$ with a purely noise-initialized MCMC. Figure 2 (b) illustrates the smooth sampling process at stage 3. We use different numbers of Langevin steps at different stages. For example, we only run MCMC with $T^{(1)} = 15$ steps at stage 1, and gradually increase the number of steps as the model grows. The maximal number of steps is 60 in our paper.

After the training, we obtain the target EBM with $E^{(S)}$. The sampling procedure by the learned model is presented in Algorithm 2. The MCMC starts from a uniform noise distribution and then runs $T^{(S)}$ steps of Langevin updates to generate samples. Note that, both the initial distribution and the number of Langevin steps are the same in both training and sampling. If we run MCMC with more steps than that used at the training stage, the oversaturation phenomenon will occur (Nijkamp et al., 2019).

---

**Algorithm 2:** CF-EBM Sampling

**Input:** model $E^{(S)}$, $T^{(S)}$ steps
**Output:** samples $\tilde{\mathbf{x}}^{(S)}$

Draw initial $\tilde{\mathbf{x}}^{(S)} \sim \mathcal{U}^{(S)}(-1, 1)$
**for** $t = 1 : T^{(S)}$ **do**
  | Update $\tilde{\mathbf{x}}^{(S)}$ by using Eq. (3)
**end**

---

### 3.3 ENERGY-BASED UNSUPERVISED IMAGE-TO-IMAGE TRANSLATION

We then generalize the proposed approach to the task of unpaired image-to-image translation and scale it up to high resolution image datasets. Given two image domains, $\mathcal{X}$ and $\mathcal{Y}$, endowing two ground-truth distributions $p(\mathbf{x}) : \mathbf{x} \in \mathcal{X}$ and $p(\mathbf{y}) : \mathbf{y} \in \mathcal{Y}$, the goal is to learn two energy functions $E_{\mathcal{X}}(\mathbf{x}) : \mathcal{X} \to \mathbb{R}$ and $E_{\mathcal{Y}}(\mathbf{y}) : \mathcal{Y} \to \mathbb{R}$, and use their short-run Langevin dynamics for image-to-image translation. Here, we only explain how to learn $E_{\mathcal{Y}}$ for one-sided image translation from $\mathcal{X}$ to $\mathcal{Y}$ since the other side is straightforward. To be specific, we train the target distribution $p(\mathbf{y}) \propto \exp(-E_{\mathcal{Y}}(\mathbf{y}))$ with the following short-run Langevin dynamics starting from the examples from source domain $\mathcal{X}$

$$\tilde{\mathbf{y}}_{t+1} = \tilde{\mathbf{y}}_t - \frac{\eta_t}{2} \nabla_{\tilde{\mathbf{y}}} E_{\mathcal{Y}}(\tilde{\mathbf{y}}_t) + \sqrt{\eta_t} \boldsymbol{\epsilon}_t, \ \boldsymbol{\epsilon}_t \sim \mathcal{N}(0, \mathbf{I}), \ \tilde{\mathbf{y}}_0 = \mathbf{x} \sim p_{\text{data}}(\mathbf{x}), \tag{4}$$

which is also our translation process. The underlying assumption of applying EBM for image translation is that the two domains $(\mathcal{X}, \mathcal{Y})$ share the same ambient space where each data can be decomposed into a content code $c$ and a domain specific style code $s_{\mathcal{X}}$ of domain $\mathcal{X}$ or $s_{\mathcal{Y}}$ of domain $\mathcal{Y}$ respectively. This assumption is

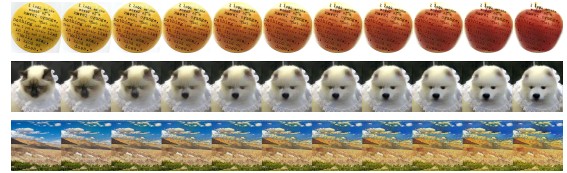

Figure 3: Evolution of energy-based image translation

somewhat similar to the partially shared latent space assumption in Huang et al. (2018); Choi et al. (2020); however, it is directly defined in the ambient space and is more efficient without resorting to complementary models. Consequently, as shown in Figure 2(c) and Figure 3, the above dynamic aims to evolve the style code $s_{\mathcal{X}}$ towards $s_{\mathcal{Y}}$, where the gradient descent steps into a lower energy area with content $c$ unchanged as much as possible.

**Generative Saliency Map:** With the proposed method, we visualize the image translation dynamics in Figure 3. Three datasets are selected for demonstration because they contain different kinds of generative saliency, *e.g.*, color, shape, and texture. To make the translation process interpretable, we adopt the gradient-based saliency maps (Simonyan et al., 2014) to highlight the essential aspects that lead to the translation, which we call *generative saliency map*. It is computed as the aggregated gradients in Eq. (4), *i.e.*, $\sum_t \nabla_{\tilde{\mathbf{y}}} E_{\mathcal{Y}}(\tilde{\mathbf{y}}_t)$. It quantifies the magnitude of the change of each pixel that contributes to the image translation. Figure 4 displays different types of generative saliency maps.

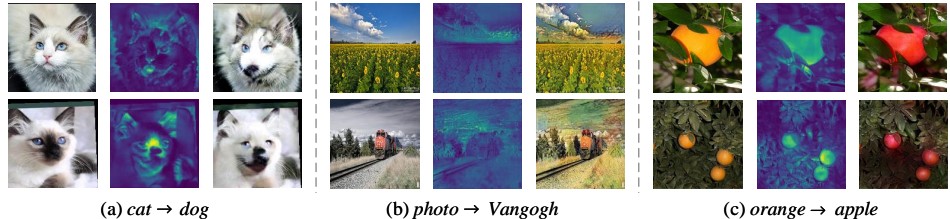

(a) *cat → dog*          (b) *photo → Vangogh*          (c) *orange → apple*

Figure 4: Unsupervised image-to-image translations with gradient-based generative saliency maps. Two examples are displayed for each of the three datasets. Each example is a triplet, including an input image, the corresponding saliency map, and its translated image, from left to right.

### 3.4 DATA PERTURBATION AND ACTIVATION FUNCTION

Most well-established EBMs add Gaussian noise to training data for stabilizing the training (Nijkamp et al., 2019; Grathwohl et al., 2020). Similarly, score-based models (Song & Ermon, 2019; 2020) inject a decayed noise to improve the score estimation. However, the additive noise will slightly change the data distribution, thus resulting in foggy synthesized examples (Grathwohl et al., 2020; Song & Ermon, 2020). See Appendix A.5 for more discussions. With the proposed coarse-to-fine expanding and sampling strategy, our EBM doesn't resort to data perturbation for stable training.

We further study the effect of the activation function used in the energy function. When the data $\mathbf{x}$ is continuous, the smoothness of the derivative of the activation function will substantially affect the Langevin sampling process (because the derivative is inside $\nabla_{\mathbf{x}} E(\mathbf{x})$ in Eq. (3)), therefore applying

an activation function with continuous gradients everywhere can stabilize the sampling. In this regard, we analyze and compare different activation functions, including ReLU (Nair & Hinton, 2010), LeakyReLU (Maas et al., 2013), CELU (Barron, 2017), Swish (Ramachandran et al., 2016), and GELU (Hendrycks & Gimpel, 2016b). We observe that the non-smooth activations like ReLU and LeakyReLU may cause a divergence of learning. See Appendix A.6 for details.

## 4 EXPERIMENTS

We showcase the results on various tasks, including image generation, image restoration (inpainting and denoising), out-of-distribution (OOD) detection and unpaired image translation. Most of them are implemented with the same network architecture and run in the TITAN X (12GB) platform. Detailed architectures and experimental setting are provided in Appendixes A.1 and A.2. We adopt Fréchet Inception Distance (FID) (Heusel et al., 2017) and Kernel Inception Distance (KID) (Bińkowski et al., 2018) for quantitative evaluation , and Amazon Turker Platform (AMT) for human perceptual evaluation (detailed in Appendix A.3). KID score is more reliable when there are fewer testing images available (*e.g.*, image translation). More synthesis results can be found in Appendix A.4. In Appendix A.6, we conduct a detailed ablation study for activation functions, normalization, etc. Finally, we scale our model up to $512 \times 512$ resolution on the CelebA-HQ dataset in Appendix A.8.

### 4.1 IMAGE GENERATION

**Datasets:** (*i*) CIFAR-10 (Krizhevsky, 2009) is a dataset containing 60k images at $32 \times 32$ resolution in 10 classes; (*ii*) CelebA (Liu et al., 2015) is a celebrity facial dataset containing over 200k images. For fair comparison with former EBM works, the $64 \times 64$ resolution is used for quantitative evaluation; (*iii*) CelebA-HQ (Karras et al., 2018) contains 30k high resolution ($512 \times 512$) facial images.

| Approach | Models | FID |
|---|---|---|
| VAE | VAE (Kingma & Welling, 2014) | 78.41 |
| Autoregressive | PixelCNN (Van den Oord et al., 2016) | 65.93 |
| | PixelIQN (Ostrovski et al., 2018) | 49.46 |
| GAN | WGAN-GP (Gulrajani et al., 2017) | 36.40 |
| | SN-GAN (Miyato et al., 2018) | 21.70 |
| | StyleGAN2-ADA (Karras et al., 2020) | **2.92** |
| Flow | Glow (Kingma & Dhariwal, 2018) | 45.99 |
| | Residual Flow (Chen et al., 2019a) | 46.37 |
| | Contrastive Flow (Gao et al., 2020) | 37.30 |
| Score-based | MDSM (Li et al., 2020) | 30.93 |
| | NCSN (Song & Ermon, 2019) | 25.32 |
| | NCK-SVGD (Chang et al., 2020) | 21.95 |
| EBM | Short-run EBM (Nijkamp et al., 2019) | 44.50 |
| | Multi-grid (Gao et al., 2018) | 40.01 |
| | EBM (ensemble) (Du & Mordatch, 2019) | 38.20 |
| | CoopNets (Xie et al., 2018b) | 33.61 |
| | EBM+VAE (Xie et al., 2021d) | 39.01 |
| | CF-EBM | 16.71 |

Table 1: FID on CIFAR-10

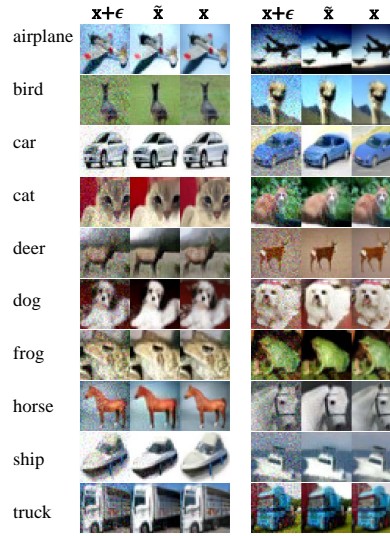

Figure 5: Image denoising

| Models | FID |
|---|---|
| VAE (Kingma & Welling, 2014) | 79.09 |
| DCGAN (Radford et al., 2016) | 32.71 |
| NCSNv2 (Song & Ermon, 2020)[†] | 28.90 |
| Short-run EBM (Nijkamp et al., 2019) | 23.02 |
| WGAN-GP (Gulrajani et al., 2017)[†] | 21.40 |
| mr-Langevin (Block et al., 2020)[†] | 19.54 |
| CF-EBM | **14.35** |
| CF-EBM[†] | **10.80** |

Table 2: FID on CelebA

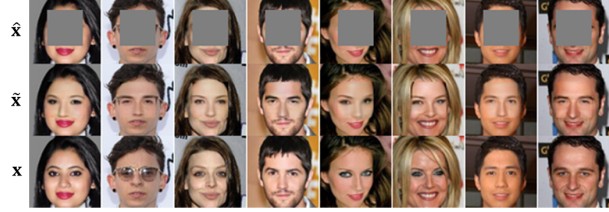

Figure 6: Image inpainting

**Quantitative Results:** The quantitative results for unconditional image generation on CIFAR-10 and CelebA are given in Table 1 and Table 2. In Table 2, the result marked with † is experimented

on the center-cropped ($140 \times 140$) and resized images of CelebA whereas others are trained on the resized CelebA. In all cases, the CF-EBM outperforms other methods by a wide margin. In Table 3, we compare our method with some representative open-sourced EBMs and the score-based model in terms of the number of parameters and the relative computational time. All of the methods rely on Langevin dynamics for sampling. We use 50 Langevin steps for sampling while all other three methods require 60, 100 and even more steps, respectively. We observe that our model consistently stands out in terms of memory and computation efficiency while keeping the lowest FID on CIFAR-10.

**Model Analysis:** Once the EBM is learned, it can be directly applied to image restoration tasks. As in Du & Mordatch (2019), we conduct two experiments on image restoration to demonstrate the model's abilities of mode coverage and generalization to unseen data. (*i*) *Image denoising*: we randomly select images from the testing dataset of CIFAR-10, and then pollute each image $\mathbf{x}$ with an additive Gaussian noise $\epsilon \sim \mathcal{N}(0, 0.2)$. Figure 5 shows the denoised results $\tilde{\mathbf{x}}$, which are obtained by Langevin dynamics initialized with $\mathbf{x} + \epsilon$, on all classes of CIFAR-10, implying a good generalization ability. (*ii*) *Image inpainting*: We mask 25% areas of each image, and use the Langevin dynamics of the learned EBM to recover the missing areas, which can be viewed as associative memory (Hopfield, 1982). As shown in Figure 6, each recovered image exhibits a meaningful but different recovered pattern than the ground truth. We also provide diverse inpainting results in Figure 11 of Appendix A.4, indicating a good mode coverage. We also provide nearest neighbors retrieved from training data for the generated images in Figure 12 of Appendix A.4 to show that our model doesn't memorize the training images but synthesizes novel ones.

| Models | #Params(M) | Time |
|---|---|---|
| EBM(ensemble) | 18.7 | 21.3 |
| NCSN | 7.6 | 14.2 |
| Short-run EBM | 2.8 | 2.2 |
| CF-EBM | **2.7** | **1.0** |

Table 3: Comparison of efficiency

**Qualitative Samples:** We have presented qualitative samples of CelebA-HQ $512 \times 512$ in Figure 1. Please refer to Appendix and see Figure 9 for CIFAR-10 samples, Figure 16 for how the noise level affects the samples' visual effect in the evaluation and Figures 17, 18 for more high resolution samples.

### 4.2 OUT-OF-DISTRIBUTION DETECTION

Out-of-distribution (OOD) detection is a binary classification problem. Du & Mordatch (2019) has shown that the likelihoods of EBMs can be useful for OOD detection. The EBM is expected to output a likelihood that is higher for an in-distribution example and lower for an out-of-distribution example. According to this principle, we conduct an OOD detection experiment and compare our model with the other three generative models, including Glow (Kingma & Dhariwal, 2018), conditional EBM (Du & Mordatch, 2019) and JEM (Grathwohl et al., 2020). All models are trained on CIFAR-10, which is considered the in-distribution dataset. We use SVHN (Netzer et al., 2011), uniform distribution (Uniform), constant distribution (Constant), interpolations of CIFAR-10 images (Interp) and CIFAR-100 (Krizhevsky, 2009) as OOD datasets. We use the Area Under the Receiver Operating Characteristics curve (AUROC) (Hendrycks & Gimpel, 2016a) as a metric for evaluation. Unlike JEM (Grathwohl et al., 2020), our method doesn't incorporate any label information at the training stage, but it still shows better or competitive results, as shown in Table 4.

| Models | SVHN | Uniform | Constant | Interp | CIFAR-100 | Average |
|---|---|---|---|---|---|---|
| Glow | .07 | 1.0 | .00 | .45 | .51 | .41 |
| EBM | .63 | 1.0 | .30 | .70 | .50 | .63 |
| JEM | **.67** | 1.0 | .51 | .65 | **.67** | .70 |
| CF-EBM (Ours) | .65 | 1.0 | **.78** | **.81** | .59 | **.77** |

Table 4: Comparison of AUROC scores in OOD detection. The higher the score, the better the performance. The other three models are all conditional generative models except for our CF-EBM.

### 4.3 UNSUPERVISED IMAGE-TO-IMAGE TRANSLATION

We use four unpaired image translation datasets for evaluation, including cat2dog, Yosemite summer2winter, vangogh2photo and apple2orange. All images are resized to $256 \times 256$ pixels. More details are provided in Appendix A.9.1. We compare our approach against CycleGAN (Zhu et al., 2017), CUT (Park et al., 2020) and two latest state-of-the-art methods U-GAT-IT (Kim et al., 2020) and FQ-GAN (Zhao et al., 2020). CycleGAN is a popular GAN-based image translation framework

and we use it to demonstrate a basic comparison between the *GAN-based model with cycle consistency constraint* and the proposed *energy-based model without cycle consistency constraint*. CUT (Park et al., 2020) is the state-of-the-art one-sided unpaired image translation framework which applies the contrastive loss to preserve the content. U-GAT-IT and FQ-GAN are current leading models on cat2dog and vangogh2photo. The baseline results are from the published papers or checkpoints.

**Quantitative Results:** Results are shown in Table 5. In most cases, our model obtains better KID scores and human perceptual scores. Meanwhile, the training time and the model size are incredibly small. The efficiencies mainly come from two aspects: *model design* and *optimization process*.

(*i*) *Model design*: The EBM only consists of an image-to-scalar energy function instead of an image-to-image mapping as in GAN-based methods. It can implicitly transfer styles and preserve contents without cycle consistency constraint by Langevin dynamics, which is guided by the energy function. Thus, the model size is much smaller than that of GAN-based model which contains a pair of GANs.

(*ii*) *Optimization process*: GANs use the adversarial loss to match styles and the cycle loss to preserve the contents. In practice, the two losses are partially decoupled and optimized alternatively. In contrast, our approach only applies MLE on one neural network (Eq. (2)) and practically, we find that the training takes less iterations to converge. Although we need to run additional Langevin steps for sampling, the overall optimization cost is still low.

| Models | $cat \rightarrow dog$ | | $dog \rightarrow cat$ | | $photo \rightarrow Vangogh$ | | $Vangogh \rightarrow photo$ | | #Params | #Days |
|---|---|---|---|---|---|---|---|---|---|---|
| | KID↓ | AMT↑ | KID↓ | AMT↑ | KID↓ | AMT↑ | KID↓ | AMT↑ | | |
| CycleGAN | 8.92 | 1.1 | 9.94 | 0.9 | 5.46 | 22.1 | 4.68 | 28.3 | 28.3M | - |
| U-GAT-IT | 7.07 | 16.0 | 8.15 | 40.9 | 4.28 | 28.7 | 5.61 | 10.5 | 671M | 6.0 |
| FQ-GAN | 6.44 | 31.3 | 8.90 | 35.3 | 6.54 | 15.9 | 5.21 | 20.4 | 671M | 6.0 |
| CF-EBM | **6.20** | **51.6** | 9.21 | 22.9 | **4.25** | **33.3** | **4.49** | **40.8** | **649K** | **0.6** |

Table 5: Quantitative results of different methods on different image translation datasets

**Qualitative Results:** As can be seen in Figure 7, our model can generate sharper images with superior visual quality than the baselines. It better preserves the source content while evolving the style from the source domain to the target domain. We put the comparison between our model and one-sided translation model CUT (Park et al., 2020) in Appendix A.9.3. More translation results on $photo \leftrightarrow Vangogh$ and Yosemite $summer \leftrightarrow winter$ can also be found in Appendix A.9.4.

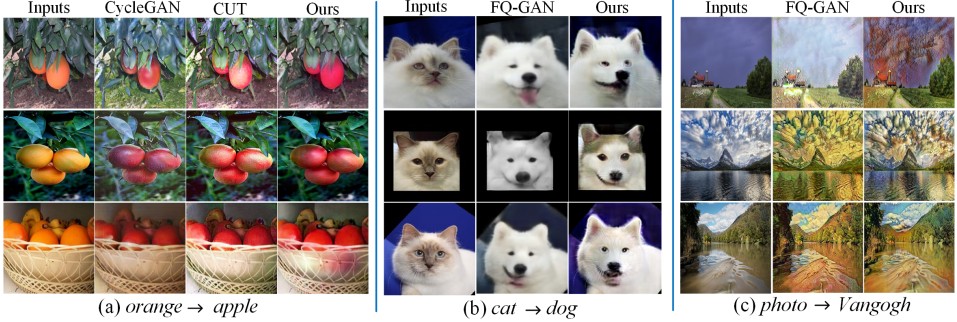

(a) *orange → apple*  (b) *cat → dog*  (c) *photo → Vangogh*

Figure 7: Qualitative comparison of different methods on different image translation datasets.

## 5 CONCLUSION

To tackle the training instability and the multimodal MCMC sampling difficulty of the EBMs, we propose a multistage coarse-to-fine expanding and sampling strategy, coined CF-EBM. Extensive experiments demonstrate the superior performance of the proposed framework, compared with various generative models, in terms of sample quality, computation and memory efficiency. The success of the CF-EBM is due to the proposed progressive expanding and sampling strategy, architecture design of energy functions, and the selective smooth activation. To the best of our knowledge, CF-EBM is the first pure EBM that can synthesize high-fidelity images and also be competent in the unsupervised image-to-image translation. Future work may include investigating the intrinsic mechanism of EBMs, using larger architectures for better performance, and exploring other energy-based applications.

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

## A  APPENDIX

### A.1  ARCHITECTURE DESIGN

The neural network architecture of the energy function for the model at $256 \times 256$ resolution is given in Table 6. Each expanding block $\mathrm{Expand}(\cdot)$ consists of a primal block $\mathrm{Primal}(\cdot)$ and a fading block $\mathrm{Fade}(\cdot)$. The $\mathrm{Primal}(\cdot)$ consists of 2 convolutional layers, where the first layer does not change the spatial size whereas the second one comes with a stride 2 for down-sampling. The kernel size is $3 \times 3$. The top $\mathrm{Primal}(\cdot)$ (without down-sampling) uses kernel size $3 \times 3$ for the first layer and kernel size $4 \times 4$ for the second layer. We also have a residual connection in $\mathrm{Primal}(\cdot)$, which is a single convolutional layer followed by an average pooling of size $2 \times 2$. We don't apply the residual connection in the last $\mathrm{Primal}(\cdot)$ block. $\mathrm{FromRGB}(\cdot)$ is a special block that transforms a 3-channel RGB image to a $z$-channel feature map, where $z$ is number of channels of each kernel at the bottom convolutional layer, in order to connect the input image to the bottom layer of the energy function. We illustrate the expansion process from resolution $16 \times 16$ to $32 \times 32$ in Figure 8.

| Block Type | Output shape |
|---|---|
| Flatten & Linear | 1 |
| Primal | $4c \times 1 \times 1$ |
| Primal | $4c \times 4 \times 4$ |
| Primal | $4c \times 8 \times 8$ |
| Primal | $4c \times 16 \times 16$ |
| Primal | $4c \times 32 \times 32$ |
| Primal | $2c \times 64 \times 64$ |
| Primal | $c \times 128 \times 128$ |
| FromRGB | $c/2 \times 256 \times 256$ |
| RGB image $\mathbf{x} \in \mathbb{R}^{3 \times 256 \times 256}$ | $3 \times 256 \times 256$ |

Table 6: The neural network architecture for energy function at resolution $256 \times 256$. $c$ is the channel multiplier and we set $c = 32$ for all experiments.

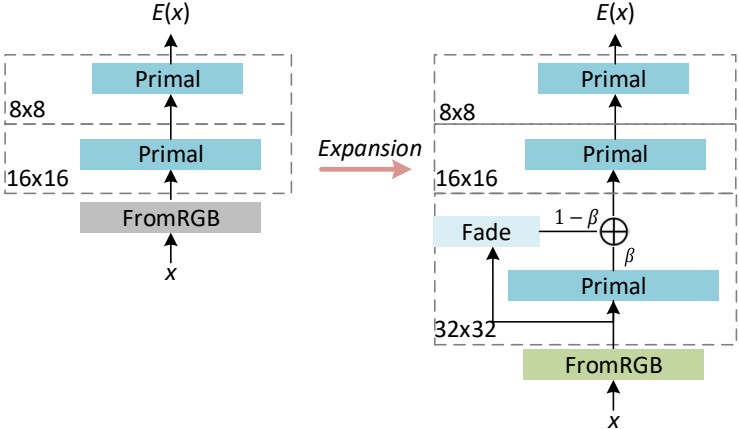

Figure 8: Illustration of the expansion from resolution $16 \times 16$ to $32 \times 32$. We first remove the FromRGB block at the bottom of the energy function at resolution $16 \times 16$, and then add an expanding block, which consists of a Primal block and a Fading block, and a new FromRGB block, which is used to connect the input RGB image to the newly added Primal block.

## A.2    EXPERIMENTAL SETTING

We train the network using Adam (Kingma & Ba, 2014) optimizer with $\beta_1 = 0.5$ and $\beta_2 = 0.999$. We set the learning rate schedule as $\alpha = \{8 \times 8 : 0.001, 16 \times 16 : 0.001, 32 \times 32 : 0.001, 64 \times 64 : 0.0012, 128 \times 128 : 0.0015\}$ and the data feeding schedule as $N = \{8 \times 8 : 50k, 16 \times 16 : 75k, 32 \times 32 : 100k, 64 \times 64 : 125k, 128 \times 128 : 150k\}$. As to the sampling hyperparameters, we set the schedule of the number of Langevin steps $T = \{8 \times 8 : 15, 16 \times 16 : 30, 32 \times 32 : 50, 64 \times 64 : 50, 128 \times 128 : 60\}$, the Langevin step size as $1.0$, and the variance of the Langevin noise term as $\eta_t = 2e^{-2} - 2e^{-2}/(T - t + 1)$ in most experiments.

We preprocess CelebA to produce CelebA-64 by using two different settings. (1) Each image is firstly center-cropped into $140 \times 140$ and then resized to $64 \times 64$. In Table 2, NCSN (Song & Ermon, 2019), mr-Langevin (Block et al., 2020), and WGAN-GP (Lee et al., 2018b) use this setting. The FID computation protocol is based on Song & Ermon (2019), where the FID score is calculated between 10k generated images and all test images. (2) Each image is directly resized to the resolution $64 \times 64$. The FID is computed on 40k generated samples. In Table 2, VAE, DCGAN, and short-run EBM follow this setting.

## A.3    EVALUATION METRICS

We consider three commonly used measures for evaluating the quality of the synthesized images.

(*i*) Fréchet Inception Distance (FID) (Heusel et al., 2017) compares the distribution of generated images with the distribution of training images. Instead of directly comparing images, the FID compares the mean and standard deviation of one of the deeper layers in Inception v3 (Szegedy et al., 2016), which is a convolutional neural network trained for classification. The FID is consistent with human perceptual judgement. Lower FID scores indicate that the model can generate images with higher quality. We follow the dataset split protocol in Zhang et al. (2020) to compute the FID.

(*ii*) Kernel Inception Distance (KID) (Bińkowski et al., 2018) is an unbiased metric that computes the squared maximum mean discrepancy (MMD) between Inception representations, making it more reliable than FID especially when the number of test images is small. For evaluating the image translation performance, we compute KID between the translated images from test images in the source domain and the test images in the target domain. Lower KID scores indicate the better translation results. We follow Kim et al. (2020) to compute KID.

(*iii*) For human perceptual evaluation using Amazon Turker Platform, we follow the setting in Zhao et al. (2020). Each testing image is judged by six participants, who are asked to select the best translated images by taking image visual quality and source content preservation into consideration. The participants are informed the information of the target domain, *e.g.*, six examples of images in the target domain are shown to them for reference.

## A.4    MORE IMAGE SYNTHESIS RESULTS

Figure 9 shows the generated images of the EBM learned from the CIFAR-10 and ImageNet-1k ($32 \times 32$ pixels) datasets. Table 7 presents the quantitative results of image generation on ImageNet-1k. The proposed CF-EBM achieves competitive results in terms of FID with much less parameters.

In Figure 10, we visualize the short-run Langevin dynamics initialized from the uniform noise distribution for generating CelebA images. Figure 11 illustrates the results of image inpainting on test images of CelebA at $64 \times 64$ resolution. Running Langevin dynamics on the occluded images, we can observe diversity of image completion. It demonstrates that our model generalizes well to unseen test data and the MCMC of the learned model has a good property of mode coverage. Figure 12 shows the generated samples and their nearest neighbors retrieved from the training data. We find that the synthesized images are not identical to the training data, which means that our model learns to generate new image patterns instead of reconstructing the existing training images.

## A.5    THE EFFECT OF DATA PERTURBATION

Almost all energy-based related generative models add Gaussian noise $\mathcal{N}(0, \sigma^2 \mathbf{I})$ to perturb the training data for stabilizing the training. We investigate the effect of such a data perturbation. Nijkamp

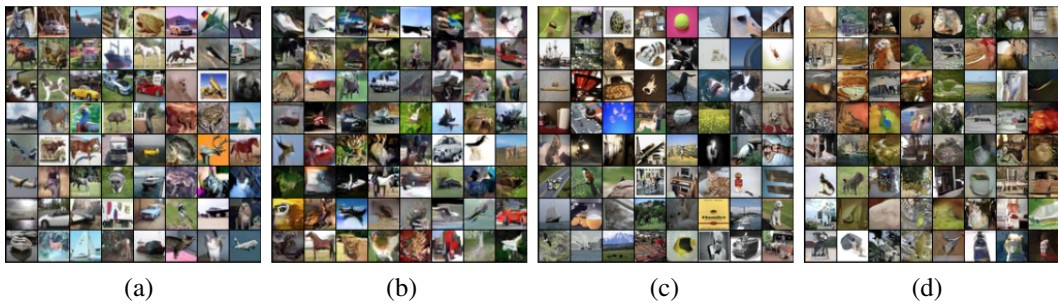

|        (a)         |        (b)         |        (c)         |        (d)         |

Figure 9: Qualitative results of image synthesis. (a) ground truth images of CIFAR-10. (b) generated images of CIFAR-10. (c) ground truth images of ImageNet-1k ($32 \times 32$ pixels). (d) generated images of ImageNet-1k ($32 \times 32$ pixels).

| Approach | Models | FID | #Params |
|----------|--------|-----|---------|
| Unconditional | PixelCNN (Van den Oord et al., 2016) | 40.51 | $> 35M$ |
| | PixelIQN-small | 37.62 | $>35M$ |
| | PixelIQN-big | 26.56 | $>50M$ |
| | CF-EBM (ours) | 26.31 | 5.82M |
| Conditional | PixelCNN | 33.27 | |
| | PixelIQN (Ostrovski et al., 2018) | 22.99 | - |
| | EBM (Du & Mordatch, 2019) | 14.31 | |
| Self-supervision | SS-GAN (Chen et al., 2019b) | 17.10 | - |
| | MS-GAN (Tran et al., 2019) | 12.30 | |

Table 7: FID on ImageNet-1k ($32 \times 32$). The channel multiplier is $ch = 48$.

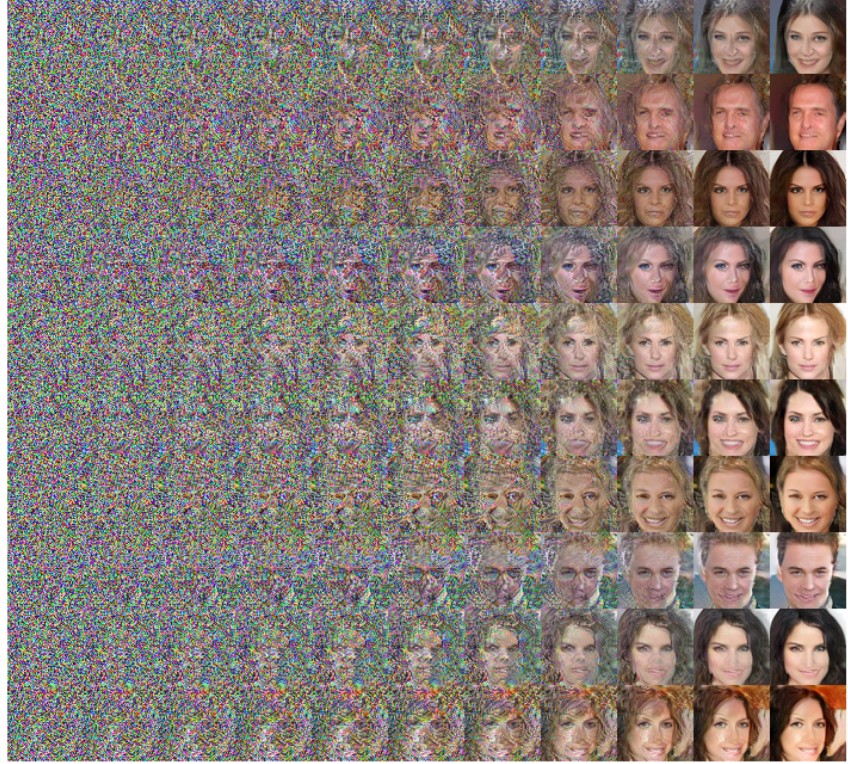

Figure 10: Illustration of short-run MCMC generative sequences on CelebA (50 Langevin steps)

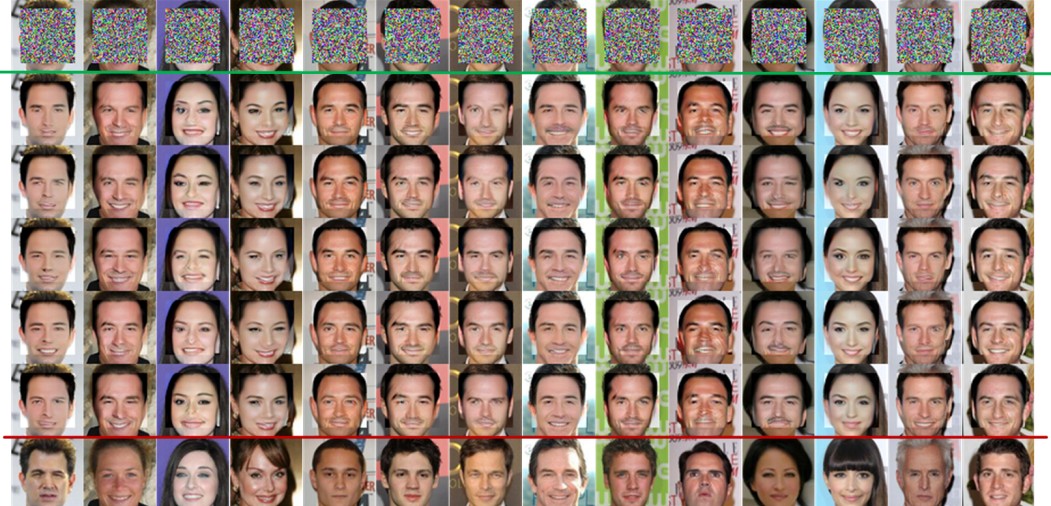

Figure 11: Illustration of diverse image inpainting. The top row shows the occluded face images and the bottom row shows the original images for reference. Other rows show different inpainting results. Each column corresponds to one example of image inpainting. Our model can complete the occluded areas with different meaningful facial expressions.

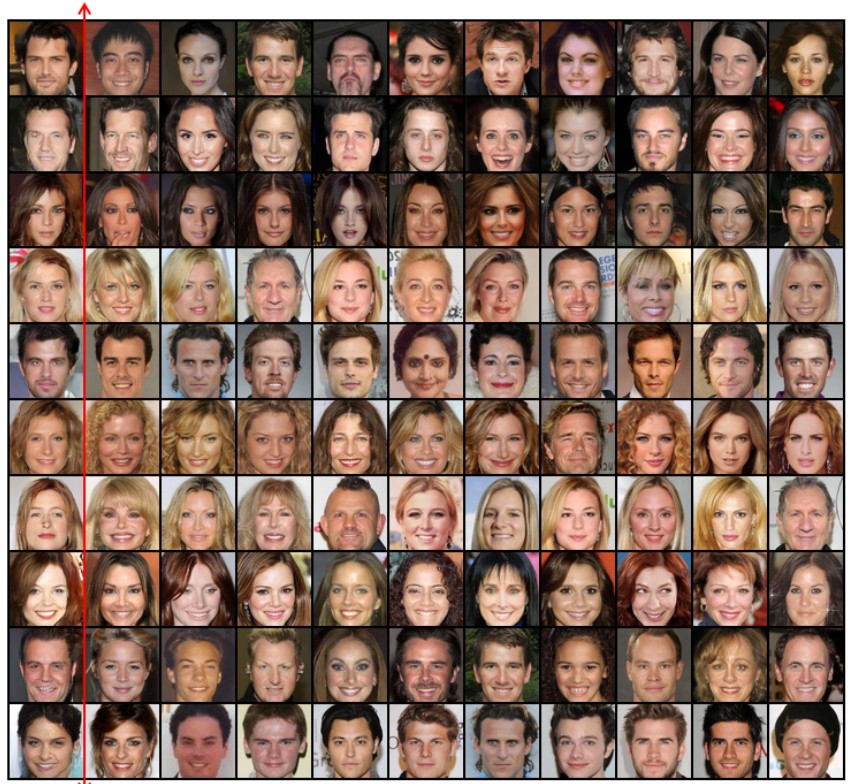

Figure 12: Nearest neighbors of synthesized images on CelebA ($64 \times 64$ pixels). Generated samples are displaced in the left column whereas their top 10 nearest neighbors retrieved from the training set are shown in the other columns.

et al. (2019) have already reported that a larger perturbation would lead to a lower FID score, and the minimum standard deviation of the additive noise they use is $\sigma = 0.03$. Also, Song & Ermon (2019) apply a perturbation with a decayed standard deviation from $\sigma = 1.0$ to 0.01. We claim that the EBM learned form the training data with additive noise will memorize the noise information even if they are very small. As a result, the synthesized images will look noisy and foggy. Figure 13 shows the effect of the data perturbation by illustrating the synthesized images generated by the models learned with different levels of data perturbations. Note that, even if the standard deviation of the additive noise is as small as $\sigma = 0.01$, the effects reflected in the synthesized images are obvious. The same observation can be found from the synthesized images shown in Figure 2 of Nijkamp et al. (2019), Figure 1 of Song & Ermon (2020) and Figure 2 of Grathwohl et al. (2020), all of which use data perturbation during training. Our model doesn't rely on the additive noise during training, thus leading to better synthesis quality.

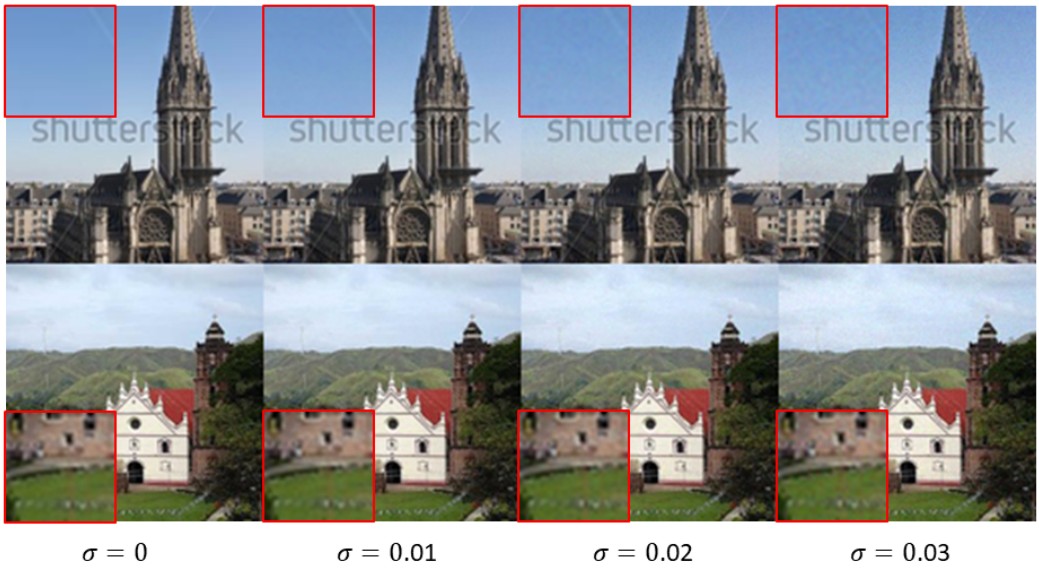

Figure 13: Comparison of different levels of data perturbations, with red rectangle areas zoomed in. Images are from LSUN (Yu et al., 2015) church at $128 \times 128$ resolution.

### A.6 ABLATION STUDY

We mainly examine the activation functions, normalization and layer connections.

#### A.6.1 ACTIVATION FUNCTIONS

We compare the following activation functions by presenting their formulas and derivatives:

- **Rectified Linear Unit** (ReLU) (Nair & Hinton, 2010): $f(x) = \max(0, x)$, and the derivative:

$$\frac{\partial f(x)}{\partial x} = \begin{cases} 1 & \text{if } x > 0 \\ 0 & \text{otherwise} \end{cases} \tag{5}$$

- **Leaky Rectified Linear Unit** (LeakyReLU) (Maas et al., 2013): $f(x) = \max(0, x) + \alpha \cdot \min(0, x)$, and the derivative:

$$\frac{\partial f(x)}{\partial x} = \begin{cases} 1 & \text{if } x > 0 \\ \alpha & \text{otherwise} \end{cases} \tag{6}$$

where $\alpha = 0.2$ is the common setting.

- **Continuously Differential Exponential Linear Unit** (CELU) ([Barron, 2017](#)): $f(x) = \max(0, x) + \min(0, \alpha \cdot (\exp(x/\alpha) - 1)$, where $\alpha$ is a shape parameter and the derivative is:

$$\frac{\partial f(x)}{\partial x} = \begin{cases} 1 & \text{if } x > 0 \\ \alpha \cdot (\exp(x/\alpha) - 1) & \text{otherwise} \end{cases} \tag{7}$$

- **Swish** ([Ramachandran et al., 2016](#)): $f(x) = x \cdot \sigma(\beta x)$, where $\sigma(\cdot)$ is Sigmoid function and $\beta$ is a learnable parameter. The derivative is:

$$\frac{\partial f(x)}{\partial x} = \beta x \cdot \sigma(x) + \sigma(\beta x)(1 - \beta x \cdot \sigma(\beta x)). \tag{8}$$

  We set $\beta = 1$ in this paper.

- **Gaussian Error Linear Unit** (GELU) ([Hendrycks & Gimpel, 2016b](#)): $f(x) = x \cdot \Phi(x)$ where $\Phi(\cdot)$ is the cumulative distribution function for Gaussian distribution. It can be approximated by $f(x) \approx 0.5x(1 + \tanh(\sqrt{2/\pi}(x + 0.044715x^3)))$.

Figure 14 visualizes the above activation functions and their derivatives. Obviously, both ReLU and LeakyReLU are non-smooth around 0. Quantitative results of image synthesis with different activation functions on CIFAR-10 are shown in Table 8. We find that models using LeakyReLU or ReLU activation function diverge early. These phenomenons are also observed in the CelebA experiment. Therefore, smoothness of the activation can improve the training stability and synthesis quality of EBMs.

| Activation function | FID |
|---|---|
| ReLU | N/A |
| LeakyReLU | 78.21 |
| GELU | 28.71 |
| CELU ($\alpha = 1$) | 21.50 |
| Swish | 16.71 |

Table 8: Effects of different activation functions

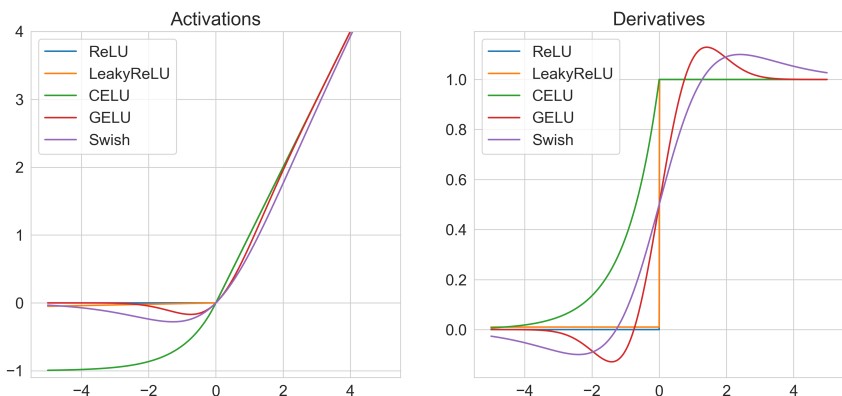

Figure 14: Visualization of different activation functions and their derivatives

### A.6.2 NORMALIZATION

Since the objective in Eq. (2) is similar to that in Wasserstein GAN ([Arjovsky et al., 2017](#)), and the bottom-up energy function in the EBM acts as a discriminator, we thus naturally consider spectral normalization ([Miyato et al., 2018](#)) to improve the performance and stability. The spectral normalization constrains the Lipschitz constant of the learnable neural network parameters, which is widely used to stabilize the training of the discriminator network of GAN.

The batch normalization is not appropriate here, because each Langevin step relies on the running mean/variance of $\mathbf{x} \sim p_{\text{data}}(\mathbf{x})$. Even if the statistics are not updated in Eq. (3), the distribution of $p_\theta(\mathbf{x}_t)$ keeps changing all the time along a chain. As a result, the pre-computed statistics of the batch normalization are not suitable for $\mathbf{x}_t$ at different step $t$. Other normalizations, *e.g.,* instance normalization and layer normalization, are not appropriate experimentally.

### A.6.3 ANALYSIS OF DIFFERENT COMPONENTS

We first examine how each component affects the model performance. They include progressive growing, residual connection, and spectral normalization. Results are shown in Table 9. From (a)-(c), we find that both spectral normalization and residual connection can improve the image synthesis quality. In (d), we disable the coarse-to-fine training but still keep residual connection and spectral normalization, the performance drops. Besides, we also compare the time efficiency regarding the training with and without the coarse-to-fine strategy. It is observed that the model without using coarse-to-fine learning requires approximately $3\times$ more time than the counterpart to converge. In (e), we sequentially train multiple EBMs at different resolutions in a coarse-to-fine manner. We try to fix the coarse EBMs when sequentially training model at higher resolution, and use the coarse EBM to initialize the sampling of the fine EBM. We keep all EBMs after training. We find that the FID of (e) is worse than that of learning a single fine EBM as in (c).

| Configuration | FID |
|---|---|
| (a) Basic CF-EBM | 32.01 |
| (b) + Spectral normalization | 20.83 |
| (c) + Residual connection | **16.71** |
| (d) – Coarse-to-fine | 24.67 |
| (e) Fix coarse EBM | 21.25 |

Table 9: Ablation study on CIFAR10.

## A.7 LIKELIHOOD EVALUATION

We conduct an experiment on the continuous MNIST for likelihood evaluation by following the setting in Du & Mordatch (2019). Its open source implementation is at https://github.com/openai/ebm_code_release. It applies the Annealed Importance Sampling (AIS) (Neal, 2001) to obtain a bound of the partition function. Figure 15 illustrates a comparison of the generated examples on MNIST between our model and the one in Du & Mordatch (2019). As can be seen, our model can produce more realistic digits. Table 10 shows a comparison of the log-likelihood among three likelihood-based models.

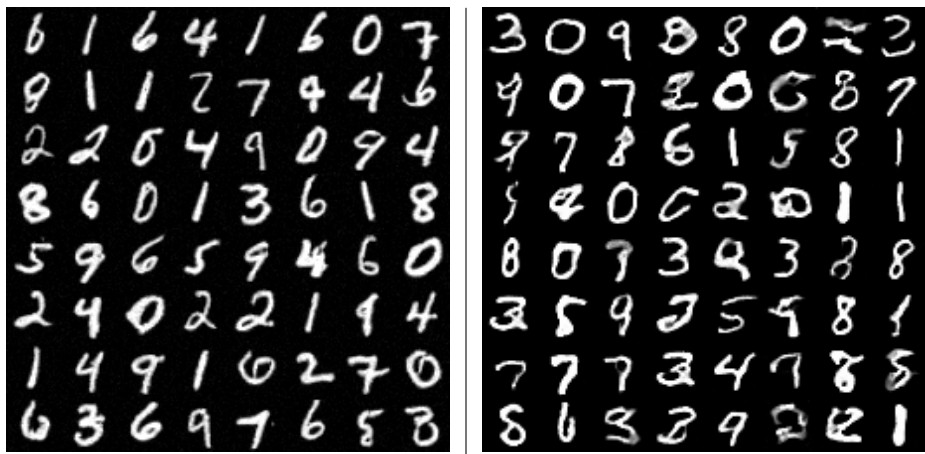

Figure 15: Image synthesis on MNIST. Left: our model. Right: EBM in Du & Mordatch (2019)

| Models | Lower Bound | Upper Bound |
|---|---|---|
| NICE (Dinh et al., 2014) | 1980.0 | 1980.0 |
| EBM (Du & Mordatch, 2019) | 1925.0 | 2218.3 |
| CF-EBM (Ours) | **2197.7** | **2342.0** |

Table 10: Log-likelihood in Nats on Continuous MNIST.

## A.8 SCALING UP THE CF-EBMS FOR HIGH RESOLUTION SYNTHESIS

To scale up EBMs for sampling high resolution images, we integrate intermediate CF-EBMs that have been learned in Algorithm 1. When the training enters into modeling higher resolution, we fix the coarse-level CF-EBM, which only produces initial points for fine-level CF-EBM samplers. In this scenario, the number of Langevin steps can be reduced to 15 to expedite sampling at each scale, meaning that generating a $256 \times 256$ images requires 90 steps. On a single TITAN X GPU (12GB), the total training time (200k iterations) of CelebA-HQ $128 \times 128$ and $256 \times 256$ are about 120 hours and 235 hours, respectively. On a single TITAN V100 GPU, these costs can be reduced to 55 hours and 100 hours (we tested it once).

In this experiment, we fix the initial variance $\eta_0 = 0.03$ for the noise term of Langevin dynamics in Eq.(3) during training. We observe that, at the testing stage, choosing a different value than that used for training will largely affect the synthetic results. As shown in Figure 16, a larger $\eta_0$ for the noise term tends to make the synthesized images not only more realistic but also more noisy. Figure 17 shows more generated samples on CelebA-HQ at $256 \times 256$ resolution. In this experiment, we set the channel multiplier $c = 32$ and train it for 5 days on a single GPU. Figure 18 shows generated samples on CelebA-HQ at $512 \times 512$ resolution where we set the channel multiplier $c = 48$ and the model is trained for 10 days on a single GPU. Table 11 compares the FID scores of different models on CelebA-HQ at $128 \times 128$ resolution. Our model obtains competitive performance without using any regularization term.

| Models | FID |
|---|---|
| Glow (Kingma & Dhariwal, 2018) | 68.93 |
| SN-GAN (Miyato et al., 2018) | 25.95 |
| WGAN-GP (Gulrajani et al., 2017) | 22.57 |
| CR-GAN (Zhang et al., 2020) | **16.97** |
| CF-EBM (ours) | 23.50 |

Table 11: FID for image generation on CelebA-HQ ($128 \times 128$). Results of GAN-based approaches are taken from Zhang et al. (2020).

## A.9 UNPAIRED IMAGE-TO-IMAGE TRANSLATION

### A.9.1 DATASET

In this paper, we experiment on five popular image-to-image translation datasets: (i) **selfie2anime** This dataset is first introduced in Kim et al. (2020), and contains a selfie domain and an anime domain, each of which has 3400 training images and 100 testing images. (ii) **photo2vangogh** This dataset is used in Zhu et al. (2017) for collection style transfer. It has 6,287 photo images and 400 Van Gogh painting images in the training set and 751 photo images and 400 Van Gogh painting images in the test set. (iii) **apple2orange** This dataset is used in Zhu et al. (2017) for object transfiguration. It has 995 training images and 266 test images for apple category and 1019 training images and 248 test images for orange category. (iv) **Yosemite summer2winter** This dataset is used in Zhu et al. (2017) for season transfer. The summer category has 1273 training images and 309 test images, while the winter category has 854 training images and 238 test images. (v) **cat2dog** This dataset is used in DRIT (Lee et al., 2018a). The numbers of images for cat and dog are 871 and 1,364, respectively. Following Kim et al. (2020), we use 100 cat images and 100 dog images as test data.

### A.9.2 EXPERIMENTAL SETTING

For the unpaired image-to-image translation task, we only made two changes compared with the setting used in image generation: (i) the channel multiplier is $c = 16$; (ii) the learning rate is multiplied by 10 only for $dog \rightarrow cat$.

### A.9.3 COMPARISON WITH CUT

We implement CUT based on its open-source code at https://github.com/taesungp/contrastive-unpaired-translation. We then select the most realistic translated results by CUT on $cat \rightarrow dog$. In Figure 19, we compare them with the proposed CF-EBM. As can be seen, CUT sometimes hallucinates a tongue, resulting in unfaithful translation results. This observation is also mentioned in the original paper (Park et al., 2020). In contrast, our CF-EBM produces more faithful translation results.

### A.9.4 MORE QUALITATIVE RESULTS ON UNPAIRED IMAGE-TO-IMAGE TRANSLATION

See Figures 20 ,21, 22, 23, 24, 25, 26, and 27 for more results of image-to-image translation.

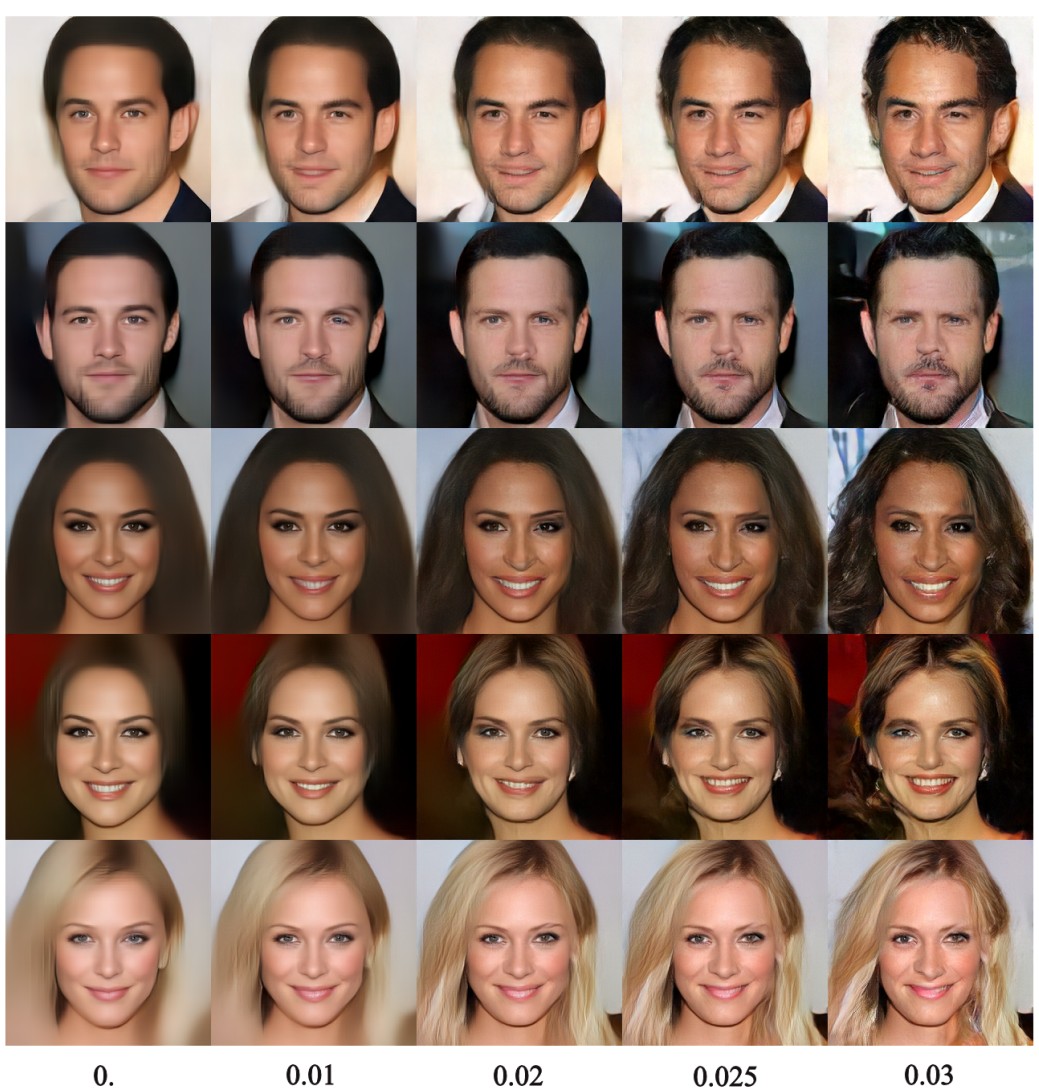

| 0. | 0.01 | 0.02 | 0.025 | 0.03 |

Figure 16: Effect of changing the initial variance $\eta_0$ for the noise term $\epsilon$ of the Langevin dynamics at the testing stage

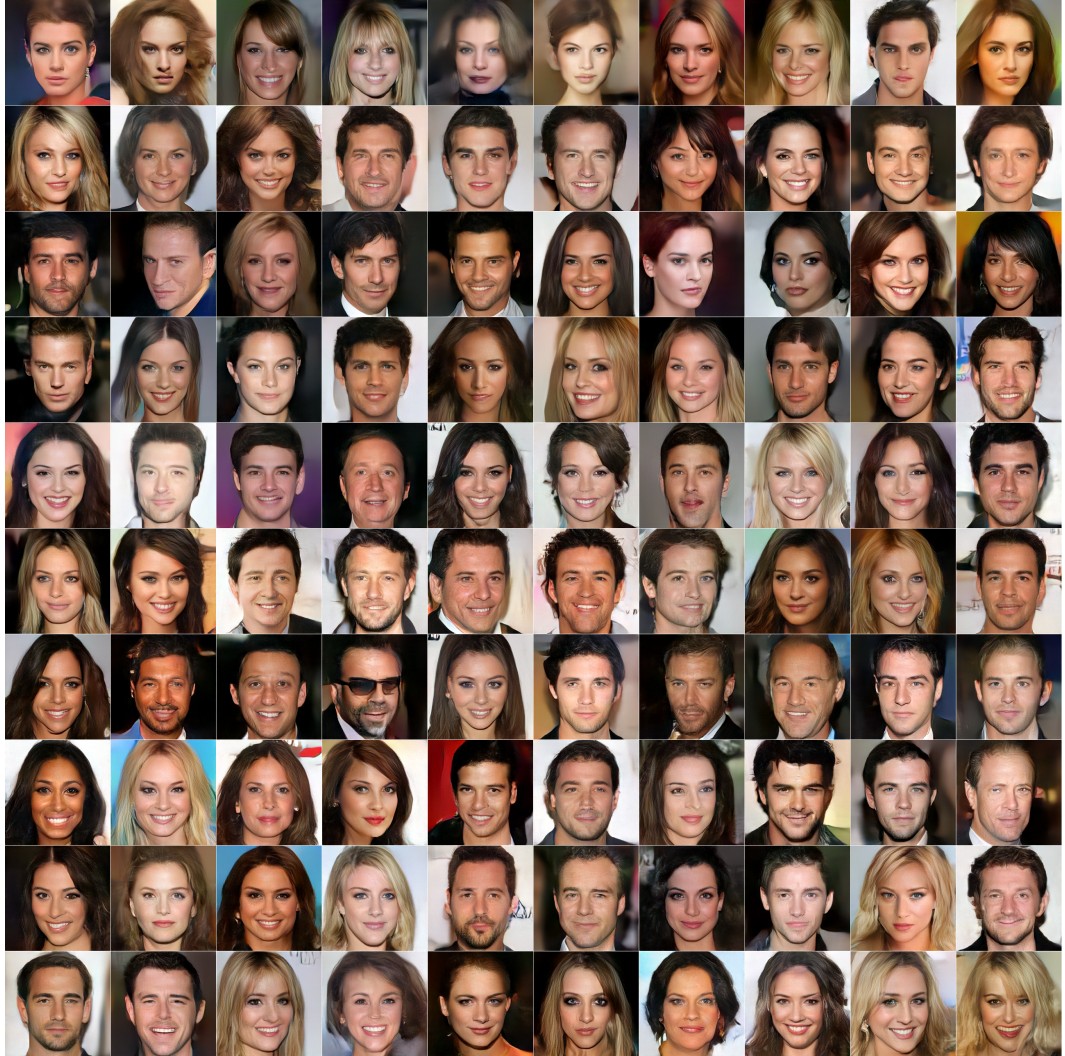

Figure 17: Generated examples on CelebA-HQ at $256 \times 256$ resolution, with a linear decay of $\eta_t$ for $\epsilon$ from 0.025 to 0.

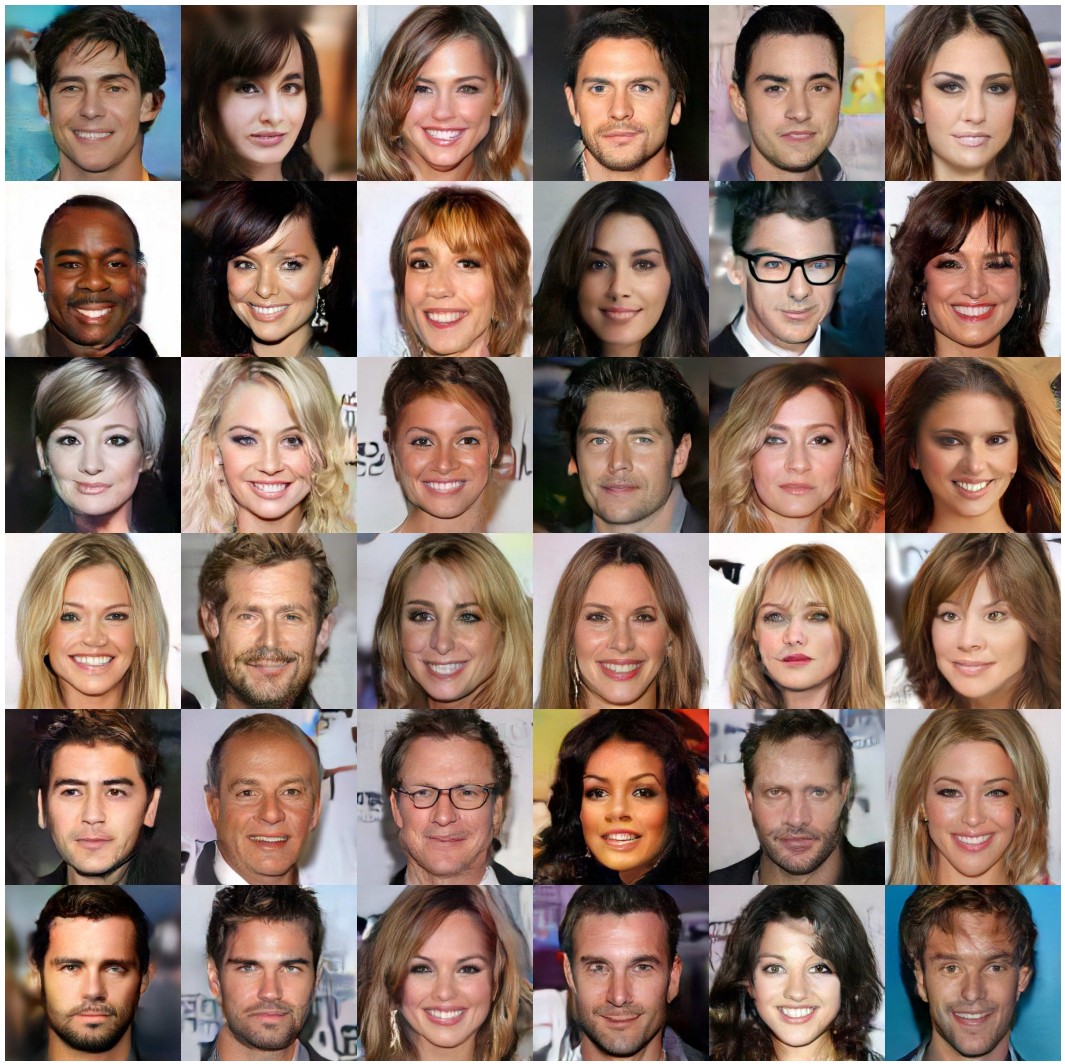

Figure 18: Generated examples on CelebA-HQ at $512 \times 512$ resolution, with a linear decay of $\eta_t$ for the noise term $\epsilon$ from 0.015 to 0.

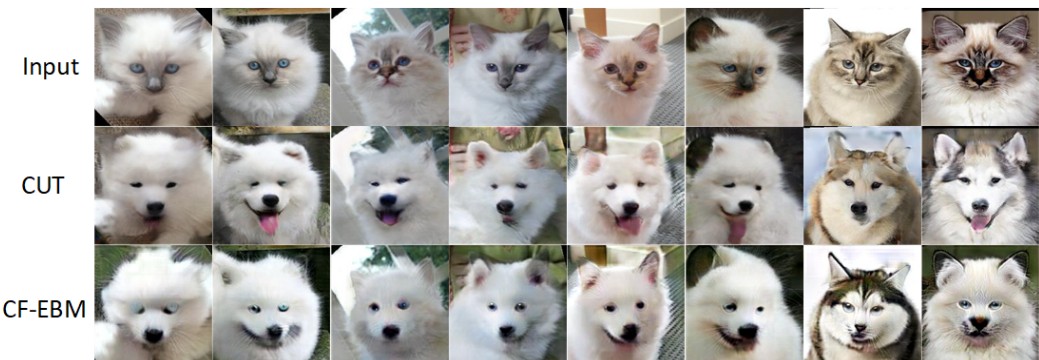

Figure 19: Qualitative comparison between our CF-EBM and CUT.

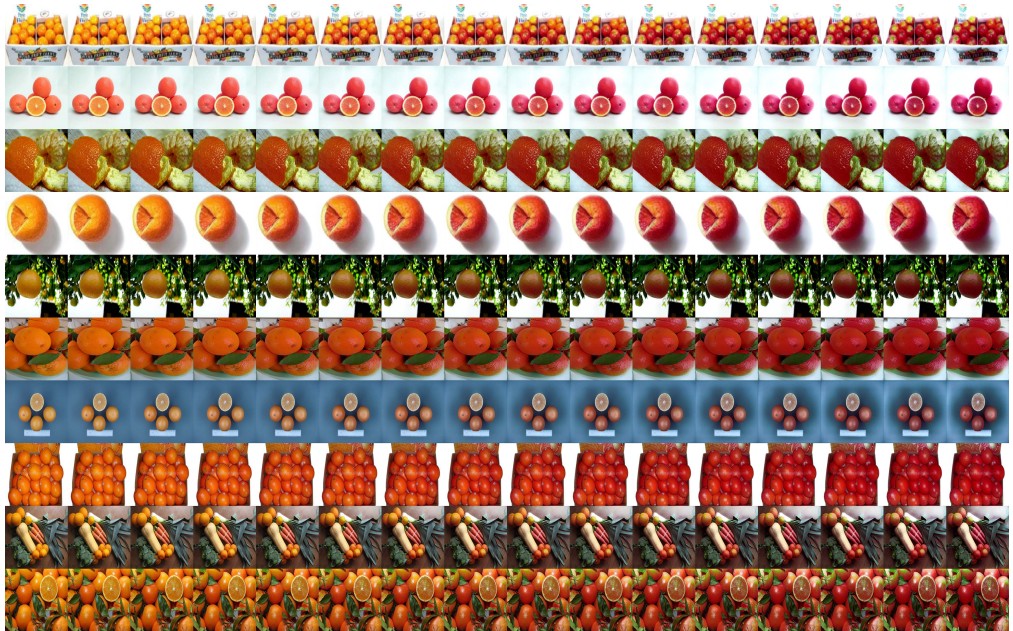

Figure 20: Translation dynamics of $orange \rightarrow apple$

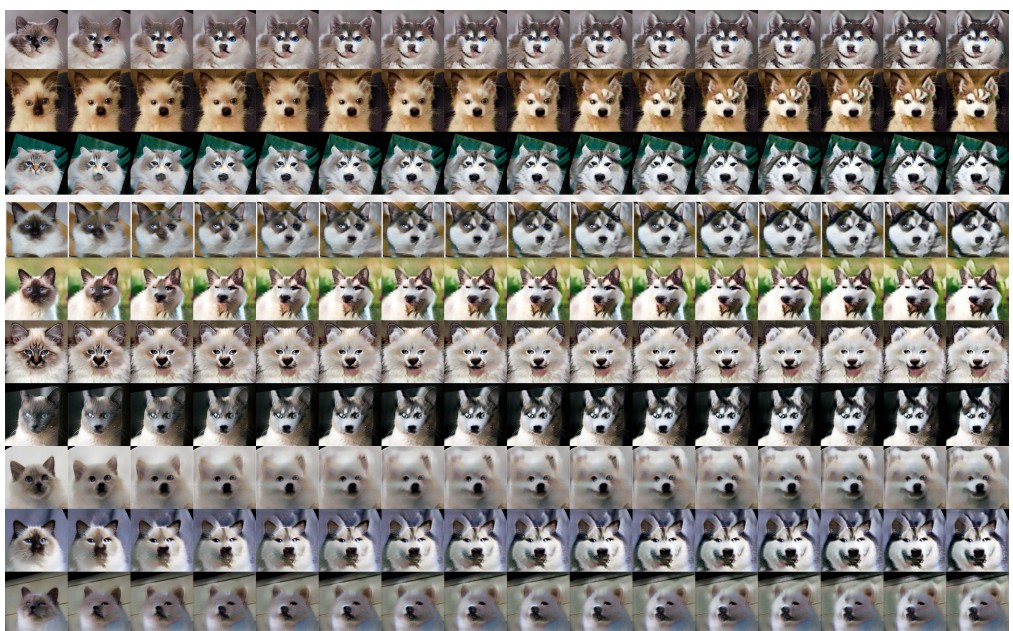

Figure 21: Translation dynamics of $cat \rightarrow dog$

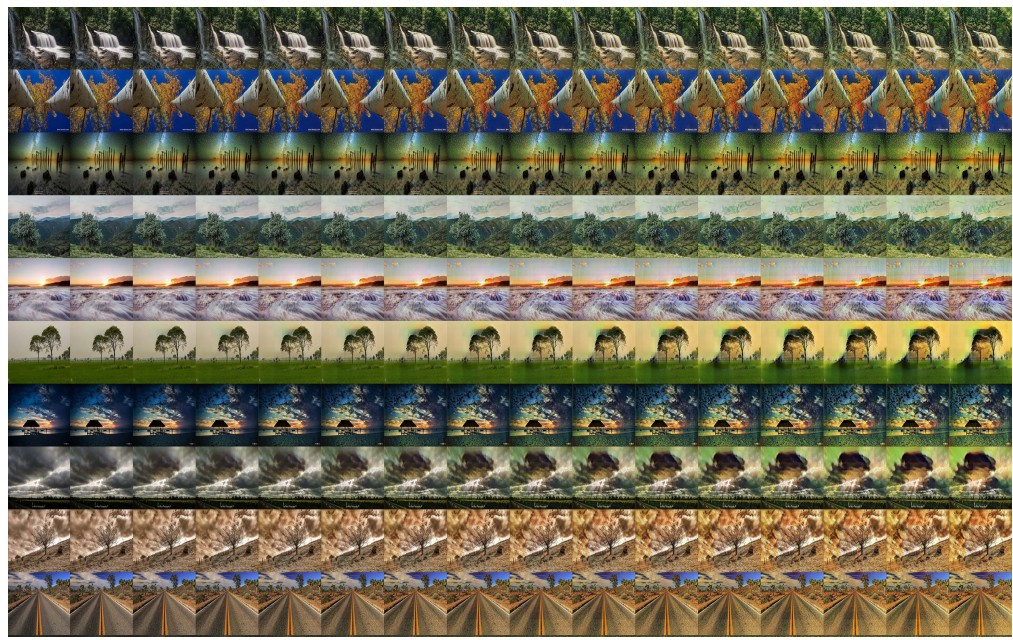

Figure 22: Translation dynamics of $photo \rightarrow vangogh$.

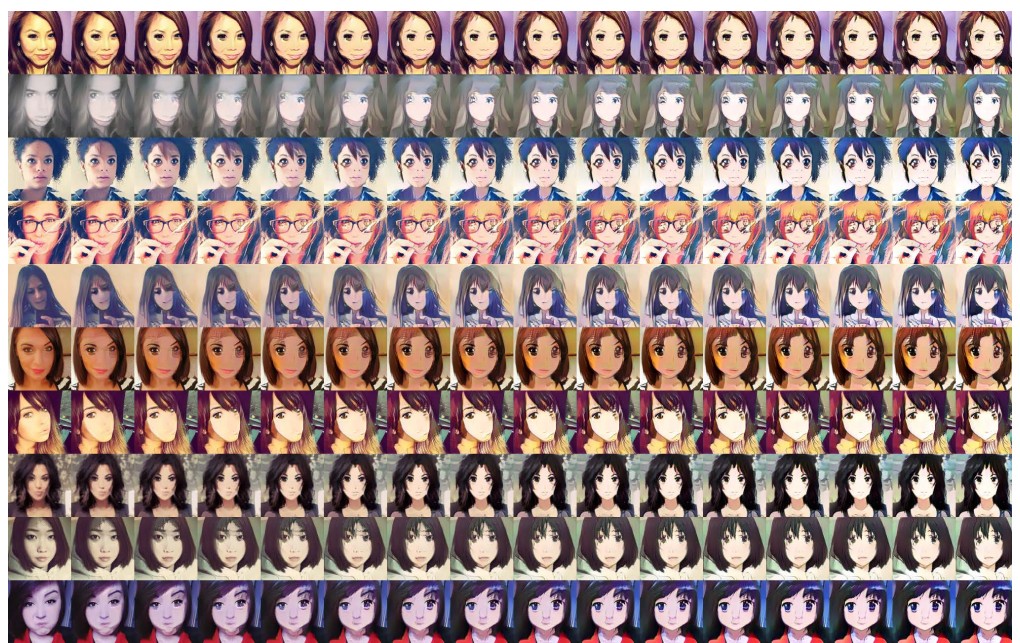

Figure 23: Translation dynamics of $selfie \rightarrow anime$

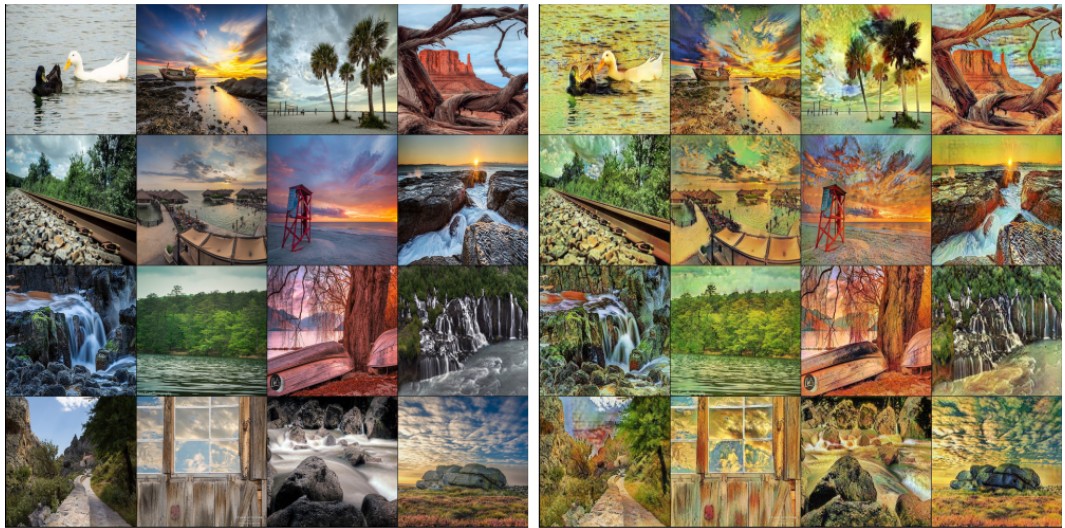

Figure 24: Translation results on $photo \rightarrow Vangogh$

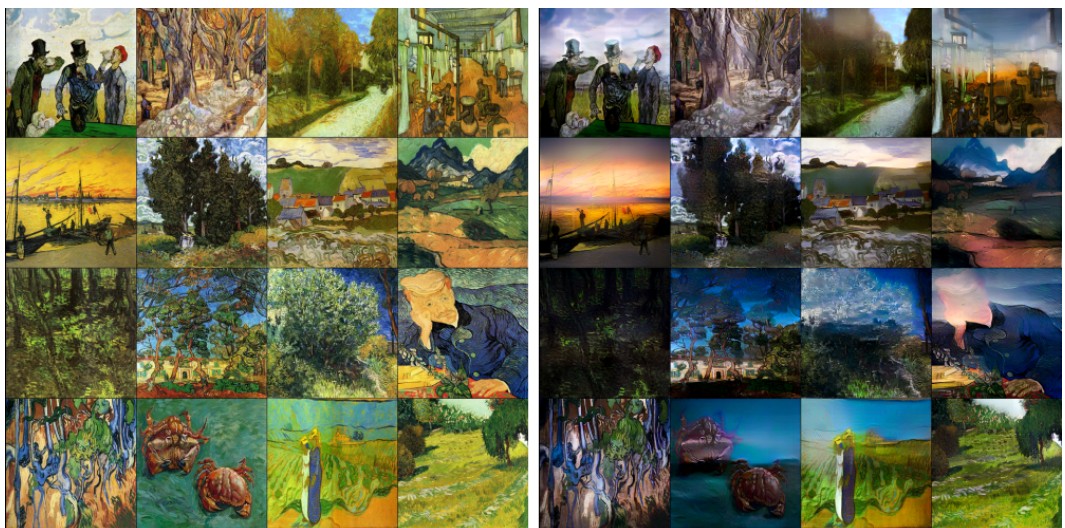

Figure 25: Translation results on $Vangogh \rightarrow photo$

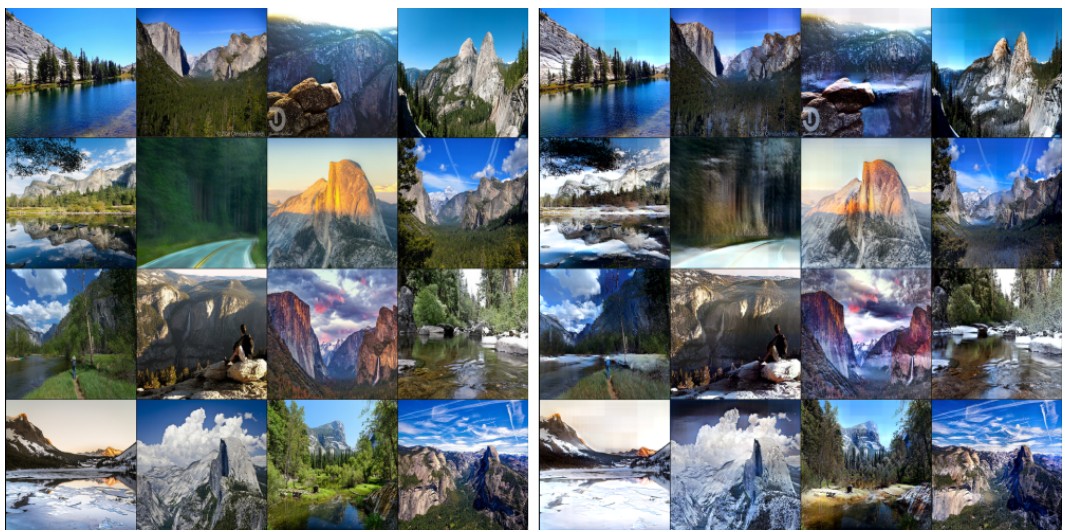

Figure 26: Translation results on Yosemite $summer \rightarrow winter$

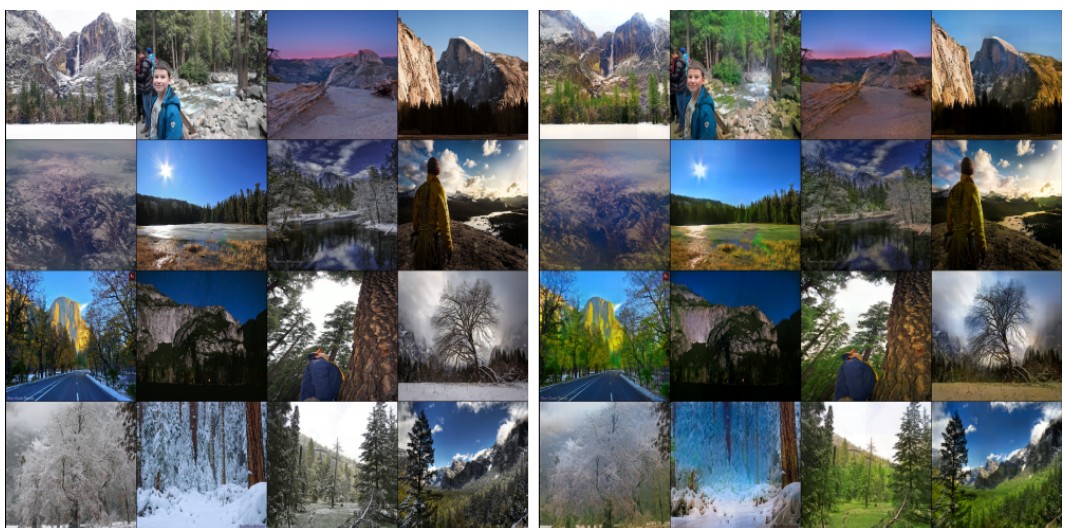

Figure 27: Translation results on Yosemite $winter \rightarrow summer$

