# OpenReview forum: "Learning Energy-Based Generative Models via Coarse-to-Fine Expanding and Sampling"
_ICLR.cc/2021/Conference — ICLR 2021 Poster_

### Official Review · AnonReviewer3 · 2020-10-25
**Simple approach, strong results, some details need clarification**

**Rating:** 7
**Confidence:** 4

**Review:**

Summary:

This paper presents a number of methods to scale up training and sampling of EBMs on image data. The main contribution consists of an approach for progressively growing the model by increasing the image resolution as training progresses. This approach echos similar approaches used for scaling up GAN training. The approach involves slowly annealing in new blocks to the model during training which processes the image at increasing resolutions. This allows training of image EBMs at notably larger resolutions than published in prior work.

The authors also present a few additional architectural improvements such as using smooth nonlinearities over the currently-popular leaky ReLU. They find that this change allows us to remove the commonly-used Gaussian noise added to the input, improving sample quality. The authors demonstrate the performance of their approach, focusing on image generation. They present results with FID and compare to other generative models and EBMs. They also examine the quality of the learned energy functions with inpainting and denoising experiments.

Beyond unconditional image generation, the authors also propose a simple method for image-translation based on EBMs. In this approach, unconditional EBMs are trained for each domain and an image is translated by taking a sample from 1 domain and running a Langevin MCMC sampler in the target domain EBM starting from the sample from the source domain. The resulting image retains many of the high level characteristics of the source image but adds low level characteristics from the target domain. Surprisingly this simple approach outperforms more involved methods based on adversarial training.

Strong areas:

The empirical results in this paper are impressive, made more impressive by the simple nature of the approach. It appears that the proposed method gives a notable improvement in image generation quality over prior EBM models while also giving an improvement in run-time and parameter-efficiency. Particularly interesting are the very strong results in the image-translation task. I am not familiar with this space so I cannot say how much of an improvement is reported here over the adversarial methods, but it is quite shocking to me that this very simple approach would outperform two recent state-of-the-art methods for this task while also having no cycle consistency objective. This should certainly bring attention to EBMs for this task.

Weaknesses:

I felt that many of the experimental details in this paper could have been made more clear. I found myself having to re-read the paper to find out that the authors used short-run MCMC instead of PCD. As these training procedures are quite different, this should be made more clear.

I also found the proposed architecture hard to understand. I could not tell if the full energy function continues to take in lower-resolution versions of the image as we add levels or if those inputs are replaced by the outputs of the added higher-resolution layers. The figures are not very clear on this. When sampling high-resolution images, do we first generate lower resolution samples, and use these to seed the high-resolution samples? Figure 2a makes it look like this is what is happening. But then algorithm 1 makes it seem like we generate samples directly from noise for the current training scale. This is important for understanding the method and it could be made much more clear. I would recommend the authors split apart algorithm 1 into an algorithm for training and one for sampling. I think that would make the method much more clear.

While the experimental results were strong, I am curious why the authors did not present any results on out-of-distribution detection (aka anomaly detection). These are common nowadays and I am curious as to how this approach will perform given its improved scalability over many EBM models which we find perform very well at this task.


My recommendation:

This is an interesting work that presents a simple method that allows image EBMs to scale up considerably. The proposed approach notably improves image generation and allows models to be trained on much larger images than has been possible by EBMs in the past. Further, the image-translation application is intriguing and the strong results should make EBMs a standard baseline for that application.

Because of these strong empirical results, I am recommending to accept this paper, but I am not advocating for a strong-accept.  While the results are strong, the work does not improve our understanding of these models, or introduce any particularly novel techniques.

I think the contributions of this work could be made greater with some experiments on some other applications such as OOD. I also think the work could be greatly improved by some rewrites to section 3.2. This is the main contribution of the paper and the authors should take more care to ensure that their effort is easily understood.

---

> ### Author Response · Authors · 2020-11-14
> **Response to Reviewer 3**
>
> Thanks for your support of our work. Here we would like to address your main concerns briefly:
>
> 1. Yes. This work is most related to short-run MCMC. The sampler always starts from a fixed uniform distribution, which is essentially different from PCD. We will make it more clear.
>
> 2. In Figure 2(a), we train the model stage by stage, and the previous low-resolution EBM is merged into the high-resolution EBM. This is consistent with Algorithm 1. Therefore, we only keep one trainable EBM. For sampling, we directly generate samples from a uniform noise, which has the same dimension as the final output image, without starting from low-resolution. We will provide a new figure and make it more evident in the revision.
>
> 3. We are working on ODD and will update it in the revision very soon.
>
> 4. Thanks for your suggestion. We are working on revising the manuscript, especially the model part. Please note the reminder of the revision, and we appreciate more advice on improving this work.

---

> > ### Author Response · Authors · 2020-11-18
> > **Revision uploaded**
> >
> > Thanks again for your support and suggestions. We have **revised the paper**, especially the model introduction, to facilitate the reading and highlight our contribution. **The OOD experiment is added in Section 4.2.** We observe that our model can obtain mostly the best or competitive results even compared with conditional generative models. Please see the detail in the uploaded revision. **We have also provided 512x512 samples in Figure 17 to make the contribution even greater.** Revised parts are highlighted in red.

---

> > > ### Comment · AnonReviewer3 · 2020-11-22
> > > **Nice improvements**
> > >
> > > I thank the authors for their thoughtful response to my review and the changes they have made. The strong OOD results are interesting and indicate that the learned energy function may be useful for tasks beyond generation. Further, the 512x512 samples are impressive and demonstrate a considerable improvement over prior EBM works.
> > >
> > > There is always value in work that pushes the boundaries of what is capable with current methods and this work has demonstrated to me that the boundaries for current EBMs were quite a bit larger than I was aware.
> > >
> > > Based on these new results and the refinements made to the paper, I will increase my score to a 7.

---

> > > > ### Author Response · Authors · 2020-11-23
> > > > **Thanks for your acknowledgement!**
> > > >
> > > > Dear Reviewer 3,
> > > >
> > > > Thanks for your acknowledgment of this work. It is very encouraging! We believe EBM generative models will bring us more surprises in the future.

---

### Official Review · AnonReviewer1 · 2020-10-28
**Energy-based generative models and progressive training**

**Rating:** 5
**Confidence:** 4

**Review:**

The authors propose an energy-based generative model for image-to-image generation that differs from previous methods by incorporating a progressive learning scheme that gradually increases the resolution of images being trained on.
Strengths:
-- The presence of a saliency map in image-to-image translation is a benefit that is harder to get out of other methods.
-- The ability to infill missing parts of an image with the same model that does unsupervised translation is a nice benefit.
-- The smoothing along resolutions during training seems like a good way to incorporate the progressive training technique into this model.
Weaknesses:
-- The novelty here is very limited. The difference between this and previous energy-based models is the scheme of progressively generating at 8x8, 16x16, 32x32, etc... which itself is a well-established technique from the Progressive GAN. Given neither of these things is new, the novelty lies in just using the one with the other.
-- The authors claim the saliency map is a benefit of their method but offer no comparisons with other models, e.g. the marginal gradient methods for any CNN model including CycleGANs.
-- The quantitative scores for the GAN methods are significantly worse than models can actually achieve now. The Big-GAN, which is several years old now and not even SOTA on CIFAR has a better score than the one proposed here.
-- The qualitative results from the GANs are much poorer than can actually be achieved. For example, the orange->apple not changing shape is a result of a poor architecture with too high of a cycle-consistency coefficient.
-- The characterization of GANs in the previous work section is very poor. The authors claim, without evidence or citation, that the invertibility of the two directions in a CycleGAN "may intensify long-standing instability issues" when it very likely does the *opposite*. They also criticize GANs for requiring an "elegant design" like instance normalization, when this is a standard and easy-to-implement part and is no more problematic than any aspects of the authors' proposed framework. Also masquerading as a "critique" is that GANs have a "fancy architecture": I don't think the authors could devise a definition of fancy that excludes their model. If this section is designed to motivate their energy-based approach, it can be done without using arbitrary, subjective, inaccurate insults of other methods.

---

> ### Author Response · Authors · 2020-11-18
> **Response to Reviewer 1 Part 1/2**
>
> We appreciate your efforts in providing constructive suggestions on the current presentation. We want to address your concerns as:
>
> **Clarification on the difference from PGAN**
> Given the fact that GAN and EBM are totally different frameworks, our proposed coarse-to-fine learning based on a single neural network is definitely non-trivial and essentially different even though EBM and PGAN share the same motivation.
>
> 1. **Different learning and sampling** PGAN [3] samples from the generator and applies a discriminator to guide the learning. In contrast, EBM integrates both learning and sampling in one neural network.
>
> 2. **The incremental learning in EBM is very difficult** In PGAN [3], the generator always learns a mapping from a **fixed** lower-dimensional prior to the higher dimensional data space. But in CF-EBM, since there is only one neural network and no such a latent variable model like generator, when we increase the resolution, the initial distribution dimension in Langevin dynamics will also **increase** accordingly. It dramatically increases the difficulty of coordinating EBM learning (Eq.2) and the sampling procedure (Eq.3). We have proposed a dedicated architecture and a smooth transition process to stabilize the incremental learning in EBM. We also demonstrate the efficiency and the better sample quality compared with other models in Table 1-3.
>
> 3. **Flexibility** Our CF-EBM is more flexible than PGAN on various image synthesis tasks. We can apply CF-EBM on image generation, image inpainting, image denoising and unpaired image-to-image translation, without changing architectures and objectives.
>
> **Choosing baselines**
> Thanks for your suggestion. In the current version, we follow previous EBM works [1, 4] that choose SNGAN as a strong baseline. We also note that there are better models on CIFAR10 generation like StyleGAN2-ADA [5]. We have added it to the revision.
>
> **About the saliency map**
> 1. **The saliency map from our method is naturally integrated so we wouldn't say it's a benefit** Because to sample from the model, we have to calculate the score function from which the saliency map is very naturally derived. Under our assumption, the score function exactly points out the magnitude and direction that the source image is adjusted to match the target domain. The saliency map is the inherent foundation that can make our approach work and provides a convenient way to visualize the dynamic translation process. Therefore, the goal of introducing the saliency map is twofold: 1) Validate our assumption that our approach can implicitly transfer styles while preserving content. 2) Give an intuitive illustration about which region has been adjusted via MCMC.
>
> 2. Indeed, we can calculate the gradient of the discriminator output in terms of the fake output from a generator in GAN-based approaches. However, this gradient won't teach the generator the exact amount and direction to edit the input to match the target but to indicate the field discriminator will respond to distinguish the fake and the real. Therefore, the gradients have different meanings between GAN and CF-EBM.

---

> > ### Author Response · Authors · 2020-11-18
> > **Response to Reviewer 1 Part 2/2**
> >
> > **About descriptions on GAN-based image-to-image translation**
> >
> > In the related work section, we want to point out the difference between them and our approach but not to criticize these works. Some words must have been misused. We have carefully revised that part and want to clarify some main points here.
> >
> > 1. **Clarification on related work** The description, the elegant design, like instance normalization and attention, means an acknowledgment. Frankly, I am a fan of GANs. These are great ideas that have been widely and successfully used, e.g. BigGAN and StyleGAN. We meant to express that the design of EBM for image translation is a little special because the MLE learning and sampling unitize the same neural network. We didn't observe performance gains, with instance normalization or attention such that the architecture is more straightforward. We have revised that part.
> >
> > 3. **Compare GAN approaches with EBM** The cycle consistency is a little restrictive, assuming a bijective mapping between two domains [2]. And, to construct the cycle consistency, we often need to optimize two generators. As shown in Figure 2(c), the proposed CF-EBM assumes that we can directly edit the source style to match the target style. The content can be preserved implicitly without explicit cycle constraints. We verify this assumption in the experiments. In summary, we solve the image-to-image translation via MCMC-based dynamic image editing while GAN learns a static image-to-image mapping.
> >
> > 5. **Clarification on experimental results** In terms of the results on orange2apple, we randomly chose from the CycleGAN website [4]. Because we found that these are visually better than the produced results from its public Google Colab [3], which we thought should be the selected checkpoint. We take this dataset as a synthetic dataset, and the only purpose of this comparison is that we want to compare the outcome of two different approaches, GAN with cycle consistency and non-GAN without the cycle consistency. Thanks for your information. We will rephrase the explanation and select the best results to make it more concrete. To be fairer, since our approach is one-sided, we have also added more results on CUT [1] in Section 4.3, which is the current SOTA for one-sided image translation using a contrastive loss to preserve the content and an adversarial loss to match the style. More comparisons can be found in Appendix A.7.2. In Table 4, the other two baselines U-GAT-IT and FQ-GAN are current SOTA on the two datasets. In contrast, our CF-EBM has fewer parameters and lower training costs while keeping competitive performances. We believe EBM is becoming an up-and-coming method in this research area.
> >
> > We have **rewritten the related work**, **add a strong GAN baseline**, **image translation experiments in Section 4.3** and **provide additional results using CUT**. **We have also provided 512x512 samples in Figure 17 to make the contribution even greater.** The revision has been uploaded and revised parts are highlighted in red. We hope your concerns are well addressed and will appreciate your time in reevaluating our work.
> >
> > [1] Yilun Du and Igor Mordatch, Implicit Generation and Generalization with EBMs, in NeurIPS 2019.
> >
> > [2] Park, Taesung, Alexei A. Efros, Richard Zhang, and Jun-Yan Zhu. "Contrastive learning for unpaired image-to-image translation.", in ECCV 2020.
> >
> > [3] https://colab.research.google.com/github/junyanz/pytorch-CycleGAN-and-pix2pix/blob/master/CycleGAN.ipynb
> >
> > [4] Grathwohl W, Wang KC, Jacobsen JH, Duvenaud D, Norouzi M, Swersky K. Your classifier is secretly an energy based model and you should treat it like one, in ICLR 2020.
> >
> > [5] https://taesung.me/cyclegan/2017/03/25/supplemental-apple-to-orange-test.html
> >
> > [6] Karras, Tero, et al. "Training generative adversarial networks with limited data.", in NeurIPS 2020.

---

### Official Review · AnonReviewer4 · 2020-10-28

**Rating:** 4
**Confidence:** 5

**Review:**

This work proposes a method to improve generation with energy based models. The work further shows how energy based models can also be extended to cross class image translation.

Strong Points:
Generated samples appear to look good

Weak Points:
My most major concern is that the overall technical novelty of the work is rather limited, and I do not think passes the bar of acceptance for ICLR. Past work [2] has shown that EBMs can composed with multi-scale sampling, while [1] notes that the smooth activation function Swish significantly improves EBM generation. Similarly, cross domain generation with EBMs have been previously demonstrated such as in [1]. While [2] uses a separate hierarchy of multi-scale EBMs, this work uses the sampling scheme in [3].

My second most major concern is other the theoretical framing of the proposed approach and well is issues with statements made in the paper. W

The proposed sampler linearly decays the noise schedule to 0, but what are the theoretical implication of such a behavior? For Langevin sampling to valid, the step size of both the model and the noise must decay to 0.

The proposed approach generates samples by running a fixed number of Langevin steps to generate a sample. But also the approach claims to train a maximum likelihood objective, where Langevin steps are used to approximate the energy landscape. A good energy landscape should be able to support an arbitrary number of MCMC sampling steps. What happens when a larger of number of steps of Langevin is run?

The paper states that past approaches such as [1] have relied on adding white noise to samples. I do not think this is the case. Can the authors elaborate?

My next major concern is that overall text of the paper is somewhat confusing to read, with some awkward sentences in the text. For example:

Section 3.2:
Assuming the total number of stages is S and the training starts from the min => Given a total of S stages of training
In what follows, be aware of the only difference => The only difference between
In this way, both stability and time efficiency in training EBMs are benefited. => In this way, both stability and time efficiency in training EBMs benefit.

Section 3.3:
We name the style that can differentiate different domains discriminative saliency. => We call this approach discriminative saliency.

Conclusion:
We owe the success to the newly proposed network architecture, smoothing activations, the powerful interpretability of EBM =>
This success is due to our newly proposed architecture, smoothing activations, and the flexibility of EBM.

To improve the clarity of the text, it would be good to perhaps add an algorithm block on the precise sampling procedure used to train EBMs. For example, the term short-run samples was never defined and I had to look at referenced papers to figure out the meaning.

The related work seems to incomplete and misses a lot of recent work towards training EBMs. Some examples of additional approaches towards training EBMs including score matching and minimum probability flow learning.

There are also some issues with the empirical evaluation of the proposed approach:

To showcase mode coverage of the EBM, it's important to show diverse inpaintings, as opposed to a single inpainting of an image.

Furthermore, as deep likelihood based model it is important to show some density/likelihood evaluation -- using some variant of AIS.

Minor Comments:

Why is the proposed approach much more efficient than past approaches for image to image translation? It seems Langevin sampling steps would take more computational time.


[1] Yilun Du and Igor Mordatch, Implicit Generation and Generalization with EBMs

[2] Ruiqi Gao, Yang Lu, Junpei Zhou, Song-Chun Zhu, and Ying Nian Wu. Learning generative convnets
via multi-grid modeling and sampling.

[3] Tero Karras et al. Progressive Growing of GANs for Improved Quality, Stability, and Variation


===== Post Rebuttal Updates

I thank the authors for responding to my comments. The work shows improved generative performance by using a multi-scale architecture, however the approach is the same that used in previous GAN works. Furthermore, the generation quality is not as good those of recent GAN works and I don't believe the added EBM benefits are significant.  In addition, other contributions, such as the use of the Swish activation and domain transfer has also been noted as used in previous work [1]. I also have additional empirical concerns over the experiments.

I list additional comments below:

1) The diverse inpaintings (Figure 11) do not really look very diverse to me and seems to suggest that the model is not learning a likelihood. To evaluate diverse inpaintings, it would be good to follow past work and evaluated on ImageNet images where only the top half of an image is conditioned. [3]

2) The likelihood evaluation are hard to interpret in A.5. An issue with evaluating AIS based likelihood sampling on MNIST for upper and lower bounds of likelihood depends heavily on the large number of steps of sampling required.  Upper and lower bounds depend heavily on the number of steps of sampling run (with unrealistically high likelihoods obtained when running only a few steps of AIS). It seems unlikely to me that the proposed model obtains a significant boost to log-likelihood compared to past approaches, and it would be good to report both the number of AIS transition distribution (and ensure that it is same used in [1]). In particular, I believe that this approach is likely to perform poorly with a large number of gradient steps (as the rebuttal response noted), which is required for proper evaluation of likelihood.

3) The related work is still missing older work in the area of EBM training. Instead of adding additional references to recent work on score based generative modeling, I think the others should cite past works that have used score matching to training energy models. For examples, such works include [4, 5, 6].

4) I didn't find the out-of-distribution results to be a particularly compelling application of the model (although its good that it performs similarly to past approaches). The only results that appear to be better are uniform (which in my experience performance across all models fluctuates) and interpolation. Furthermore, the model from [1] used in section 4.2 is not conditional.

5) I wouldn't say this paper is the first to generate 512x512 samples with EBMs. For example see [3].

6) It's difficult to evaluate an open source code release, since the code is not provided at the time.

7) Regarding the approach in [1], when doing source translation images are initialized from ground truth images from a seperate class.


[1] Yilun Du and Igor Mordatch, Implicit Generation and Generalization with EBMs

[2] Aaron van den Oord, Nal Kalchbrenner, Koray Kavukcuoglu, Pixel Recurrent Neural Networks

[3] Tian Han et al. Divergence Triangle for Joint Training of Generator Model, Energy-based Model, and Inferential Model

[4] Jascha Sohl-Dickstein et al. Minimum Probability Flow Learning

[5] Saeed Saremi. Deep Energy Estimator Networks

[6] Hyvärinen, Aapo. Estimation of non-normalized statistical models by score matching.

---

> ### Author Response · Authors · 2020-11-15
> **Novelty Clarification Part 1/2**
>
> Thanks for your insightful feedback on our presentation and suggestions for improvements. We will first address your concern about the novelty of our paper.
>
> #### Clarification on the difference from MultiGrid
> Our work essentially differs from MultiGrid [2] in the following three aspects:
>
> 1. **Different modeling strategy** MultiGrid creates multiple EBMs (i.e., multiple energy networks) to model different resolutions simultaneously, while our CF-EBM only has one single EBM, which grows gradually. Our model always keeps one learnable EBM.
>
> 2. **Different training strategies** The two models have totally different training strategies. To be specific, in MultiGrid, each EBM needs a separate optimization and a separate sampling chain in each iteration. Each EBM uses its own synthesized examples to update its energy function. It requires a careful balance to learn and design these EBMs because the lower resolution EBM needs to produce samples to initialize the MCMC of the subsequent higher resolution EBM. Since MultiGrid trains all EBMs together, the whole framework will fail if one EBM collapses. The MultiGrid learning scheme appeared to be difficult when we tried to scale up. Our work trains only one EBM progressively and runs only one MCMC chain in each iteration. We propose a smooth transition and a dedicated architecture for EBM such that the training is very stable.
>
> 3. **Different empirical analysis** Compared with MultiGrid, our method has presented a much higher resolution and better qualitative image samples and demonstrated a promising application on the unpaired image-to-image translation. In Table 1-3, the proposed model achieves the best performance among all prior EBM models including MultiGrid, while using the fewest parameters and fewest sampling steps. We have shown **256x256** samples, which are firstly seen in EBM generative models. We also prepare to add **512x512** samples to demonstrate our model's scalability in the revision. Section 4.2 presents the superior performance even compared with notable GAN approaches to the image translation task.
>
> #### Clarification on the difference from PGAN
> Given the fact that GAN and EBM are totally different frameworks, our proposed coarse-to-fine learning based on a single neural network is definitely non-trivial and essentially different even though EBM and PGAN share the same motivation.
>
> 1. **Different learning and sampling** [3] samples from the generator and applies a discriminator to guide the learning. In contrast, EBM integrates both learning and sampling in one neural network.
>
> 2. ** Incremental learning in EBM is very difficult**. In [3], the generator always learns a mapping from a **fixed** lower-dimensional prior to the higher dimensional data space. But in CF-EBM, since there is only one neural network and no such a latent variable model like generator, when we increase the resolution, the initial distribution dimension in Langevin dynamics will also **increase** accordingly. It dramatically increases the difficulty of coordinating EBM learning (Eq.2) and the sampling procedure (Eq.3). We have proposed a dedicated architecture and a smooth transition process to stabilize the incremental learning in EBM. We also demonstrate the efficiency and the better sample quality compared with other models in Table 1-3.
>
> 3. **Flexibility** Our CF-EBM is more flexible than PGAN on various image synthesis tasks. We can apply CF-EBM on image generation, image inpainting, image denoising, and unpaired image-to-image translation, without changing architectures and objectives.

---

> > ### Author Response · Authors · 2020-11-15
> > **Novelty Clarification Part 2/2**
> >
> > #### About Swish activation
> > **Our work and Du et al. [1] claimed essentially different aspects of choosing activations. Swish doesn't equal to smooth activations.**
> > [1] suggests the use of Swish to increase stability but has not claimed that we can obtain better sample quality. Note that, in the appendix of [1], it says the default activation is LeakyReLU throughout the architecture. In our paper, we notice that all kinds of **smooth activations** can increase training stability and improve performance, like GELU and CELU, not limited to Swish. We have conducted ablation experiments on how to choose activations in A.4.1. We also find that if we smooth the data with a bigger noise, all activations won't suffer from training instability. As shown in Table 7, only smooth activations can stabilize the training when we disable the data perturbation.
> >
> > #### Clarification on cross-domain sampling
> > **Du et al. [1] and our work are essentially different**
> >
> > **Du et al. [1]**: It first trains a conditional EBM, and then in the testing stage, it uses an image but conditioned on a different class to initialize the Langevin dynamics. Note that, Du et al. [1] applies a replay buffer where we get the initial samples most of the time during training. It improves over persistent contrastive divergence (PCD).
> >
> > **Ours**: As to our image-to-image translation, we train the model from scratch. The sampling process is the same both in training and testing. Given that the short-run MCMC [7] can be considered as a flow-based generator model, thus in our learning scheme, we learn a short-run MCMC that maps from the source domain to the target domain, therefore the trained short-run MCMC can serve as a translator in testing. Because  [1] doesn't use images from the source domains to initialize the short-run MCMC, their MCMC in the testing stage can not be used as a valid translator.
> >
> > **Experimental quality comparison** As shown in Figures 4, 7, and 17-24 of our paper, our model can effectively preserve source content while transferring the style, which is the ultimate goal in image-to-image translation. In contrast, [1] doesn't give very competitive examples as can be seen from Figure 6 and Figure 19 in [1].
> >
> > Last but not least, our EBM-based translation strategy is one significant contribution of our paper, demonstrating that short-run MCMC of EBM [7] can be used for unpaired image-to-image translation, which is also more efficient than GAN-based methods, as shown in Section 4.2. Intriguingly, a non-GAN method can offer a promising future in this direction.
> >
> > [1] Yilun Du and Igor Mordatch, Implicit Generation and Generalization with EBMs, in NeurIPS 2019.
> >
> > [2] Ruiqi Gao, Yang Lu, Junpei Zhou, Song-Chun Zhu, and Ying Nian Wu. Learning generative convnets via multi-grid modeling and sampling, in CVPR 2018.
> >
> > [3] Tero Karras et al. Progressive Growing of GANs for Improved Quality, Stability, and Variation, in ICLR 2018.
> >
> > [4] Welling M, Teh YW. Bayesian learning via stochastic gradient Langevin dynamics, in ICML 2011.
> >
> > [5] Grathwohl W, Wang KC, Jacobsen JH, Duvenaud D, Norouzi M, Swersky K. Your classifier is secretly an energy based model and you should treat it like one, in ICLR 2020.
> >
> > [6] Song Y, Ermon S. Improved techniques for training score-based generative models, in NeurIPS 2020.
> >
> > [7] Nijkamp E, Hill M, Zhu SC, Wu YN. Learning non-convergent non-persistent short-run MCMC toward energy-based model, in NeurIPS 2019.
> >
> > [8] Song, Yang, and Stefano Ermon. "Generative modeling by estimating gradients of the data distribution.", in NeurIPS 2019.

---

> ### Author Response · Authors · 2020-11-18
> **Address Remaining Concerns Part 1/2**
>
> **About your concern on Langevin dynamics**
> 1. **Theory and Practice** In Bayesian sampling [4], both the learning rate and the noise should be decayed to 0 to ensure convergence theoretically. However, in CF-EBM, we run a short-run MCMC [7], a non-convergent, non-mixing, and non-persistent MCMC to sample realistic images. Short-run essentially means we always run a fixed number of MCMC steps and always start from a fixed distribution. The papers [5] also elaborates that we are not running a valid SGLD sampler in practice; however, this approximate and biased sampler is still a valid generator, which enables us to generate realistic images and allows for faster training as seen in most well-established EBM works [2, 5, 7]. Due to limited space, please refer to [7] which has a very detailed discussion on this. Therefore, it won't be an issue to be taken as a generative model.
>
> 2. **Langevin steps** If the MCMC is evaluated with more steps than used in training, the over-saturation will happen [7] because the short-run MCMC is not mixing. We have also tried other Langevin steps (30-100) for training and found that 50 steps can give us the best performance in terms of FID and training time.
>
> **About more related works**
>
> Strictly speaking, score matching models [6, 8] don't count as EBMs. Because they directly learn the score function, which is the gradient of the energy function, instead of learning the energy function as in EBMs. That's why we exclude them from the EBM related work section. However, considering they also use Langevin dynamics for sampling, we have added some discussions in the revision.
>
> **About the likelihood evaluation**
>
> It is a pretty good suggestion for further improvement. We have considered two workarounds to evaluate the model likelihood:
>
> 1. **Estimate the partition function**: Follow the work [1], we conducted an experiment on continuous MNIST. It is observed that our model could generate more realistic digits compared with [1] and obtain the best log-likelihood among all models. We present the generated images and a detailed comparison in Appendix A.5 of the revision.
>
> 2. **A task with a direct link to likelihood**: The out-of-distribution (OOD) detection directly applies a density model (energy function in our case) to fit the data and produce a score. The score will be higher for in-distribution data than out-distribution data. [1, 5] have shown that the likelihood of EBM is a reliable detector. Besides, we use the area under the receiver-operating curve (AUROC) to compare models, which is derived as the relative likelihood of in-distribution and out-distribution samples. Therefore, in Section 4.2 of the revision, we have conducted an experiment on OOD and demonstrated an improvement over various likelihood models.
>
> In summary, our model demonstrates a better MLE-based learning outcome.
>
> **About the white noise injection**
> 1. **Reference** The right references here (Section 2.1) should be [5] and [7], which match the descriptions in Section 3.4 and A.3.1. Thanks for pointing this out.
>
> 2. **Adding noise eases training** Both the short-run [7] and the work JEM [5] add a Gaussian noise with a standard deviation of 0.03 to the training data. In [7], Section 3.1  claims that the injected noise makes it easy for convergence, and Section 5.4 discusses the influence of injected noise. We also tried the short-run without adding noises and found that it diverged at an early stage. [5] also made similar statements in the paper, and we observed the diverged training around epoch 3 based on their open-source code if no noise was added. The score-based models [6, 8] don't belong to EBM generative models. They are another clever work-around and state that the added noise is important to improve score estimation under the manifold hypothesis (Section 3 and Section 5 in [6]). So in our paper, we concluded based on these facts.
>
> However, the additive noise can be remembered by samplers. With the proposed learning scheme and the improved architecture, we can train EBM stably without adding noises to violate the observed data.

---

> > ### Author Response · Authors · 2020-11-18
> > **Address Remaining Concerns Part 2/2**
> >
> > **About the efficiency of CF-EBM on image translation**
> > We want to clarify two main points:
> >
> > 1. **Efficient model design**  Image-to-image translation aims to preserve the source content while translating the source style to the target style. We assume that EBM could implicitly transform the style while implicitly preserving the content. The saliency map in Figure 4 verifies our assumption. Therefore, the image-to-scalar EBM can use a very small model to edit pixels via MCMC directly. In contrast, the generator in GAN is an image-to-image mapping for translation. The model size is usually very large to transfer styles while preserving content. As shown in Table 4, our model has much fewer parameters than GANs.
> >
> > 2. **Efficient optimization** GANs require adversarial learning to match the style and cycle consistency to preserve the content. The adversarial loss optimizes both discriminators and generators. In the meantime, the cycle loss optimizes the generators. In this sense, the style matching and content preservation are decoupled. However, EBM only includes learning one neural network that implicitly preserves the content and simultaneously transfers the styles. We find that EBM can find the path to transfer styles in fewer iterations.
> >
> > In summary, EBM has fewer parameters to optimize and requires much fewer iterations to translate two domains. The EBM training cost will be low, as shown in Table 4, though there are additional Langevin steps.
> >
> >
> > **What to expect in the revision:**
> >
> > 1. **Model Introduction**: we have rewritten the model introduction (Section 3.2) and add a sampling subsection (Section 3.2.2) to facilitate the reading.
> >
> > 2. **Short-run Description**: we add a block to describe the short-run at the end of Section 3.1.
> >
> > 3. **Likelihood Evaluation**: we add an experiment of likelihood evaluation in Appendix A.5.
> >
> > 4. **Out-of-distribution detection**: we add the OOD experiment in Section 4.2.
> >
> > 5. **Related work**: we add more related work like score matching models in Section 2.1
> >
> > 6. **Diverse inpaintings**: we add diverse inpaintings in Figure 11.
> >
> > 7. **512x512 samples**: we add 512x512 samples in Figure 17 to make our work more distinct and the contribution greater.
> >
> > Revised parts are highlighted in red. We hope your concerns are well addressed and will appreciate your time in reevaluating our work.

---

> ### Author Response · Authors · 2020-11-23
> **Discussions**
>
> Dear Reviewer 4,
>
> Kindly let us know if our response below addressed your concerns. We will be happy to respond if there are additional questions. Thanks.

---

### Official Review · AnonReviewer2 · 2020-10-29
**Nice work on up-scaling EBMs and using them for unsupervised image translation**

**Rating:** 6
**Confidence:** 4

**Review:**

Much like progressive growing of GANs two years ago, this paper adopts a similar coarse-to-fine procedure for scaling EBMs to higher resolutions. In particular, the approach starts from learning EBMs on low-resolution images and then smoothly transitions to higher resolution by carefully designing an expand layer and a smooth sampling procedure. Authors were able to obtain competitive FID scores on CIFAR-10, and demonstrate the first set of 256x256 image samples from EBMs. In addition, authors demonstrate successful application of EBMs to unpaired image-to-image translation.

#### Pros
* Strong experimental results. The FID scores obtained are among the best achievable results with maximum likelihood training. Scaling to 256x256 images is a great advancement of the field. Various experiments on denoising and inpainting demonstrates the versatile applications of EBMs.

* Applications in unpaired image-to-image translation demonstrate strong potential of the approach in image editing

* Sampling typically takes 50 Langevin steps. This is much faster than denoising score matching with Langevin dynamics or denoising diffusion probabilistic models, which typically need thousands of Langevin steps.

#### Cons

* FID computation is done using a PyTorch implementation. Although the results should be close to the original FID implementation based on TensorFlow, they are not exactly comparable to previous results. The numbers in tables are therefore not rigorous.

* There is no widely agreed protocol on how to compute FIDs on CelebA 64. The processing methods can be quite different in different papers, and the number of samples used for FID computation can also be different. Please include your settings explicitly in the paper, otherwise the comparison in the table is not that meaningful.

* The progressive growing procedure requires multiple stages of separate training and also need careful tuning of model architecture and sampling to make transition smooth.

* There is a factual error in Appendix A.3.1.The work of Song & Ermon (2020) does not apply a sequence of decayed perturbation from 1 to 0.01. Quite the reverse, starting from noise scales different from 1 is one main point in that paper.

-------------
Post-rebuttal

Reporting PyTorch-computed FIDs risk the fairness when comparing against previous work. The repo quoted by the authors had a well-known issue, leading to much lower numbers compared to those computed by the original TensorFlow repo. Although this issue has been alleviated following some code update this year, the numbers still won't exactly match those of TensorFlow. Therefore, from a scientific perspective, the FID numbers have to be recomputed with the original TensorFlow code for a rigorous publication.

In addition, the FID number reported by the authors for their model is computed between 10k samples and the test dataset, while most other FID numbers in Table 1 are computed between 50k samples and the training set (following the original settings of [1]). This also makes the comparison not fair.

I completely agree that FID is a flawed metric and lower FID scores do not necessarily indicate better sample quality. However, since the authors choose to report FIDs and compare against those directly ported from previous work, they have to follow the convention and ensure a fair comparison. Since I didn't get a satisfying response from the authors (authors claimed to "have corrected it", they just changed phrasing of their response but didn't re-compute scores), I decide to lower my scores from 7 to 6. I am still marginally inclined to acceptance, but will leave it to the discretion of the AC on whether this paper should be rejected due to flawed FID computation.

[1] Heusel, M., Ramsauer, H., Unterthiner, T., Nessler, B. and Hochreiter, S., 2017. Gans trained by a two time-scale update rule converge to a local nash equilibrium. In Advances in neural information processing systems (pp. 6626-6637).

---

> ### Author Response · Authors · 2020-11-14
> **Response to Reviewer 2**
>
> Thanks for your acknowledgment of our work! Your concerns are addressed as:
>
> 1.  **FID computation** To give a fair FID comparison between Tensorflow and PyTorch implementations, the adopted repository is most widely used by researchers in PyTorch community, *e.g.* in short-run EBM [1].
>
> 2. **CelebA FID comparison** We preprocess CelebA to produce CelebA-64 in two settings. 1) Each image is firstly center-cropped into $140\times140$ and then resized to $64\times64$. In Table 2, three baselines NCSN, mr-Langevin [3], and WGAN-GP applied this setting. The FID computation protocol is also based on NCSN where the score is calculated between 10k samples and all test images. 2) Each image is directly resized to $64\times64$. FID is computed on 40k generated samples. In Table 2, VAE, DCGAN, and short-run EBM exactly follow this setting. We will clarify these in the Appendix.
>
> 3. **Model design and training** The proposed multi-stage learning scheme in EBM is very efficient with a smaller model size and fewer sampling steps. We list the model comparison in Table 3. The current model is very easy to scale up with one single GPU. In contrast, some EBMs with very deep networks require multiple GPU for training. We will also release the code and contribute to the community. To facilitate the use, everyone can play with this model and only need to change the channel number of the CNN block or import a customized data loader.
>
> 4.  **Clarification of Song \& Ermon (2020)** Regarding the description of Song \& Ermon (2020), thanks for pointing this out. The reference should be Song \& Ermon (2019). We have corrected it.
>
> Thanks again for your insightful reviews, and we will include these suggestions in the revision soon. Please note the reminder of the upcoming revision.
>
> [1] Nijkamp E, Hill M, Zhu SC, Wu YN. Learning non-convergent non-persistent short-run MCMC toward energy-based model, in NeurIPS 2019.
>
> [2] Song, Yang, and Stefano Ermon. "Generative modeling by estimating gradients of the data distribution.", in NeurIPS 2019.
>
> [3] Block, Adam, et al. "Fast Mixing of Multi-Scale Langevin Dynamics under the Manifold Hypothesis.", 2020.

---

> > ### Author Response · Authors · 2020-11-18
> > **More suggestions**
> >
> > We have uploaded the revision. Please feel free to give more suggestions!

---

> > ### Comment · AnonReviewer2 · 2020-11-23
> > **FID computation**
> >
> > Thanks for the response. For NCSN, I think the authors used the tensorflow code for computing FIDs. See the footnote of Page 15 in https://arxiv.org/pdf/1907.05600.pdf

---

> > > ### Author Response · Authors · 2020-11-24
> > > **Thanks for the information!**
> > >
> > >  We have corrected it.

---

### Public Comment · ~Zhisheng_Xiao1 · 2020-11-12
**Some questions regarding the experiments**

Hi authors. It is a great work that provide a nice way to effectively scale up the training of EBMs. I have a few questions regarding the experimental results.

1. In your ablation table 8, is line (d) meaning to train an EBM (with SN and residual connection) directly at 32*32 scale? If yes, is it with the same setting as in [1]? Their network also has SN and residual connection, but the FID result (40 or so) is much worse than your 24.67. Could you please clarify what makes this ablation baseline much better than [1]?

2. The network structure in table 5 is for images of size 256 or 128. Do you use similar design for CIFAR? If yes, then the network is similar to that of [1].

3. Do you use persistent training (replay buffer) as in [1]?

[1]Implicit Generation and Modeling with Energy Based Models. Du et al.

---

> ### Author Response · Authors · 2020-11-12
> **Some Clarifications on the Experiments**
>
> Thanks for your comments!
> 1. We use quite different architectures from [1]. For example, we have different designs of residual blocks, including details of kernels, strides, activations (smooth whereas the default in [1] is lrelu), and initializations (kaiming_normal for conv block); the spectral normalization is not used in every block. As shown in Table 3, the two models have quite different numbers of parameters in unconditional CIFAR10 generation. So at $32\times32$, the network in [1] is not the baseline that we report in Table 8. We will provide more details in the revision and release the code ASAP.
> 2. Please refer to the first response. For CIFAR10 generation, basically, you can abandon the blocks that have a higher resolution than $32\times32$.
> 3. No. We directly sample from the uniform noise and run a fixed number of Langevin steps.

---

> > ### Public Comment · ~Zhisheng_Xiao1 · 2020-11-12
> > **Cool!**
> >
> > Thanks for the quick reply! I took a look at table 3 and related text. So you design an EBM architecture with much fewer parameters, needs fewer LD steps, does not need reply buffer, while obtain significantly better results than [1]. This is really cool and it can stand out alone even w/o the coarse-to-fine techniques. Can't wait for the code release!

---

### Author Response · Authors · 2020-11-23
**General Response To All Reviewers**

Dear Reviewers,

We would like to thank all reviewers for their constructive feedback and suggestions on improving this work! We have uploaded a revision and the revised parts are highlighted in red. The updates are summarized as:

**More Experiments**
To consolidate this work, we add more experiments in response to all reviewers
1. Add **out-of-distribution detection** experiment

2. Add **$512\times512$**-resolution samples

3. Add a **likelihood evaluation**

4. Add **a comparison with CUT** which is a newly proposed one-sided unpaired image-translation model

5. Add **diverse inpaintings**

**Polished Writing**
We have polished the work to clarify our contribution and facilitate the reading.

1. Revise **related works** and add baselines in response to Reviewer 4 and 1

2.  Add **description on short-run** and **clarification on Langevin dynamics** for EBM generative model in response to Reviewer 4

3.  Revise **the model description and the algorithm** to facilitate the reading

4. Add **justifications for GAN-based and EBM-based image translation** in response to Reviewer 4 and 1

We hope our pointwise responses below could address all reviewers’ concerns. We thank all reviewers again.

---

### Comment · ~Petr_Mokrov1 · 2023-02-02
**Open-sourcing the code**

Dear authors. Could you please open-source the code of your proposed model.

---

> ### Author Response · Authors · 2023-02-03
> **link for the code**
>
> https://github.com/YangNaruto/ProEBM

---

### Decision · Program_Chairs · 2021-01-07
**Final Decision**

**Decision:**

Accept (Poster)

**Comment:**

This work proposes to train EBMs using multi-stage sampling. The EBMs are then used for generating high dimensional images, performing image to image translation, and out-of-distribution detection. The reviewers are impressed with the results, but indicate that the novelty is limited. While I agree that the work can be seen as a combination of previously proposed techniques, demonstrating that this combination can be made to work well is still a significant contribution to the field. In addition, the paper demonstrates strong results in using Langevin dynamics to translate between images, which I do think is novel. I therefore recommend accepting the paper for a poster presentation.